# The *Acinetobacter baumannii* Mla system and glycerophospholipid transport to the outer membrane

Cassandra Kamischke[1], Junping Fan[1], Julien Bergeron[2,3], Hemantha D Kulasekara[1], Zachary D Dalebroux[1], Anika Burrell[2], Justin M Kollman[2], Samuel I Miller[1,4,5]*

[1]Department of Microbiology, University of Washington, Seattle, United States; [2]Department of Biochemistry, University of Washington, Seattle, United States; [3]Department of Molecular Biology and Biotechnology, The University of Sheffield, Sheffield, United Kingdom; [4]Department of Genome Sciences, University of Washington, Seattle, United States; [5]Department of Medicine, University of Washington, Seattle, United States

**Abstract** The outer membrane (OM) of Gram-negative bacteria serves as a selective permeability barrier that allows entry of essential nutrients while excluding toxic compounds, including antibiotics. The OM is asymmetric and contains an outer leaflet of lipopolysaccharides (LPS) or lipooligosaccharides (LOS) and an inner leaflet of glycerophospholipids (GPL). We screened *Acinetobacter baumannii* transposon mutants and identified a number of mutants with OM defects, including an ABC transporter system homologous to the Mla system in *E. coli.* We further show that this opportunistic, antibiotic-resistant pathogen uses this multicomponent protein complex and ATP hydrolysis at the inner membrane to promote GPL export to the OM. The broad conservation of the Mla system in Gram-negative bacteria suggests the system may play a conserved role in OM biogenesis. The importance of the Mla system to *Acinetobacter baumannii* OM integrity and antibiotic sensitivity suggests that its components may serve as new antimicrobial therapeutic targets.
DOI: https://doi.org/10.7554/eLife.40171.001

*For correspondence:
millersi@uw.edu

**Competing interests:** The authors declare that no competing interests exist.

## Introduction

Gram-negative bacteria are enveloped by two lipid bilayers, separated by an aqueous periplasmic space containing a peptidoglycan cell wall. The inner membrane (IM) is a symmetric bilayer of glycerophospholipids (GPL), of which zwitterionic phosphatidylethanalomine (PE), acidic phosphatidylglycerol (PG), and cardiolipin (CL) are among the most widely distributed in bacteria (*Zhang and Rock, 2008*). In contrast, the outer membrane (OM) is largely asymmetric and composed of an inner leaflet of GPL and an outer leaflet of lipopolysaccharide (LPS) or lipooligosaccharide (LOS) (*Pelletier et al., 2013*). The OM forms the first line of defense against antimicrobials by functioning as a molecular permeability barrier. The asymmetric nature of its lipid bilayer and the structure of LPS/LOS molecules facilitates barrier function, as the core region of LPS impedes the entry of hydrophobic molecules into the cell while the acyl chains within the bilayer also help to prevent the entry of many hydrophilic compounds (*Bishop, 2014*). Although progress has been made in understanding many aspects of OM assembly – including the discovery of an LPS transport system and the machinery for proper folding and insertion of outer membrane proteins (*Okuda and Tokuda, 2011*; *Okuda et al., 2016*) – little is known about the molecular mechanisms for transport of the GPLs necessary for OM formation and barrier function.

**eLife digest** Gram-negative bacteria are a large group of single-celled organisms that share a typical external envelope. This casing is formed of an inner and an outer membrane, which have different structures and properties.

The outer membrane lets nutrients penetrate inside the cell, but blocks out other compounds, such as antibiotics. It is made of a complex assembly of molecules, including glycerolphospholipids (GPL) that are produced inside the cells. Very little is known about how this external shield is created and maintained. For example, it was still unclear how GPLs were exported through the inner membrane to the outer one.

To investigate these questions, Kamischke et al. exposed a species of Gram-negative bacteria to a molecule that is normally blocked by the outer membrane. If the outer membrane is not working properly, the compound can cross it and the cell turns blue.

Kamischke et al. then introduced genetic changes at random locations in the genomes of the bacteria. If colonies became blue, this meant that the mutations had happened in a gene essential for the outer membrane. Sequencing these blue bacteria revealed 58 genes involved in keeping the outer membrane working properly. Amongst them, four genes coded for a transport machine, the Mla system, which allowed GPLs to cross the inner membrane and reach the outer membrane. The experiments also showed that a working Mla system was required for bacteria to survive antibiotics.

Certain dangerous Gram-negative bacteria are now resistant to many drugs, having evolved unique envelopes that keep antibiotics at bay. By learning more about the outer membrane, we may be able to create new treatments to bypass or to disable this shield, for example by targeting the Mla system.

DOI: https://doi.org/10.7554/eLife.40171.002

*Acinetobacter baumannii* is an important cause of antibiotic-resistant opportunistic infections and has significant innate resistance to disinfectants and antibiotics. Similar to other Gram-negative opportunistic pathogens such as *Pseudomonas aeruginosa* and *Klebsiella* spp., individuals with breached skin or damaged respiratory tract mucosa are most vulnerable (*Chmelnitsky et al., 2013*; *Abbo et al., 2005*). We performed a genetic screen to identify genes important for the OM barrier of *A. baumannii*. This led to the identification of an ABC (ATP-binding cassette) transporter complex that promotes GPL export to the OM. Transporter disruption attenuates bacterial OM barrier function, resulting in increased susceptibility of *A. baumannii* to a wide variety of antibiotics.

The homologous system for *E. coli* has previously been termed Mla for its suggested role in the maintenance of outer membrane lipid asymmetry via the removal of GPL from the outer leaflet of the OM to the IM. While this is a reasonable hypothesis, there is not direct biochemical evidence that the Mla system functions to return GPL from the OM to the IM. In this work, we present evidence that the *A. baumannii* Mla system functions to promote GPL movement from the IM to the OM. This conclusion is based on the observation that newly synthesized GPLs accumulate at the IM of *mla* mutants, akin to how LPS molecules accumulate at the inner membrane in bacteria with mutations in the *lpt* genes encoding the LPS ABC transport system (*Okuda et al., 2016*). Given the broad conservation of Mla in prokaryotic diderm organisms, the anterograde trafficking function of Mla might be exploited by a variety of species.

## Results

### A screen for activity of a periplasmic phosphatase identifies genes required for *A. baumannii* OM barrier function

We identified strains with mutations in genes required for maintenance of the *Acinetobacter baumannii* OM barrier by screening transposon mutants for the development of a blue colony phenotype on agar plates containing the chromogenic substrate BCIP-Toluidine (XP). Although *A. baumannii* carries an endogenous periplasmic phosphatase enzyme, colonies remain white on agar plates containing XP. We reasoned that lesions in genes necessary for the OM barrier function should result in a blue colony phenotype, as the XP substrate becomes accessible to the periplasmic

enzyme (*Strauch and Beckwith, 1988*; *Lopes et al., 1972*). Screening roughly 80,000 transposon-containing colonies for the blue colony phenotype yielded 364 blue colonies whose insertions were mapped to 58 unique genes (*Supplementary file 1*). We confirmed the results of the screen by assaying for OM-barrier defects using ethidium bromide (EtBr) and N-Phenyl-1-naphthylamine (NPN) uptake assays (*Helander and Mattila-Sandholm, 2000*; *Murata et al., 2007*). We also tested for resistance to antimicrobials, including trimethoprim, rifampicin, imipenem, carbenicillin, amikacin, gentamicin, tetracycline, polymyxin B, and erythromycin. Greater than 85% of the strains identified in the screen demonstrated decreased OM barrier function compared to wild type. Out of the 58 strains with transposon insertions, 23 demonstrated OM permeability defects by NPN and EtBr uptake assays, and 49 out of 58 resulted in increased sensitivity to two or more antibiotics compared to the parent strain, indicating that the screen identified lesions causing OM barrier defects leading to increased permeability to small charged and hydrophobic molecules, including commonly used antibiotics.

## The Mla system is necessary for *A. baumannii* OM integrity

Four mutants with a blue colony phenotype contained unique transposon insertions in the genetic loci A1S_3103 and A1S_3102, predicted to encode core components (*mlaF* and *mlaE*) of a multicomponent ABC transport system. These genes are within a five-gene operon that encodes for a conserved proteobacterial ABC transport system homologous to the *E. coli mla* system previously implicated in OM integrity (*Malinverni and Silhavy, 2009*). The *A. baumannii* operon includes: *mlaF* and *mlaE,* respectively predicted to encode the nucleotide-binding and transmembrane domains of an ABC transporter; *mlaD,* encoding a protein containing an IM-spanning domain and a predicted periplasmic soluble domain; *mlaC,* encoding a soluble periplasmic protein; and *mlaB,* an additional gene predicted to encode a cytoplasmic sulfate transporter and anti-sigma factor antiagonist (STAS)-domain protein (*Figure 1A*). An additional putative OM lipoprotein is encoded on *mlaA*, or *vacJ,* which is clustered with the rest of the *mla* operon in some Gram-negative bacteria, although it is at a different chromosomal location in *A. baumannii*. MlaA has been functionally associated with the rest of the Mla components in *E. coli*, as mutations in *mlaA* yield comparable phenotypes to mutations in other components of the system (*Strauch and Beckwith, 1988*).

Bioinformatic analysis predicts that the *mlaC* and *mlaF* genes respectively encode the soluble periplasmic component and cytoplasmic ATPase component of the ABC transport system, and we chose to focus on mutants of these genes for further experiments to elucidate the function of the *mlaFEDCB* operon. Chromosomal deletions were created by allelic exchange, and these mutations resulted in OM permeability defects as measured by EtBr uptake assays. We complemented the OM defect for the Δ*mlaC* and Δ*mlaF* deletion mutants by repairing the original deletion event in the chromosome and confirmed complementation of the observed permeability defect (*Figure 1B*). Deletions in *mlaF* and *mlaC* also rendered *A. baumannii* increasingly sensitive to a variety of antibiotics as determined by MIC measurements (*Figure 1D*). Increased sensitivity to antibiotics whose uptake is not mediated by OM porins is consistent with a direct effect on the membrane component of the OM permeability barrier (*Nikaido, 2003*; *Vaara, 1992*). In addition to OM defects, the *mla* mutants display phenotypes that may correlate with OM stress, including increased production of extracellular carbohydrates as evidenced by crystal violet staining of pellicles following growth in broth culture (*Figure 1—figure supplement 1A*). These data indicate a role for Mla in the maintenance of the outer membrane barrier of *A. baumannii*.

## ATPase activity of MlaF is required for maintenance of the OM barrier of *A. baumannii*

To exclude the possibility that the membrane defect was the result of the disruptive effect of a partially formed Mla protein complex, we engineered an enzymatically inactive ATPase component and expressed the defective enzyme from a plasmid. We reasoned that by expressing this allele in the wild type bacteria we could create a dominant-negative effect on Mla function. The cytoplasmic ATPase component of the Mla system, MlaF, contains the consensus sequence GxxxxGKT at residues 49–56, characteristic of a Walker A motif. Downstream residues 173–178 contain the sequence LIMYDE, typical of a Walker B motif. The Walker motifs form highly conserved structures critical for nucleotide binding and hydrolysis (*Walker et al., 1982*). The lysine residue of the Walker A motif is

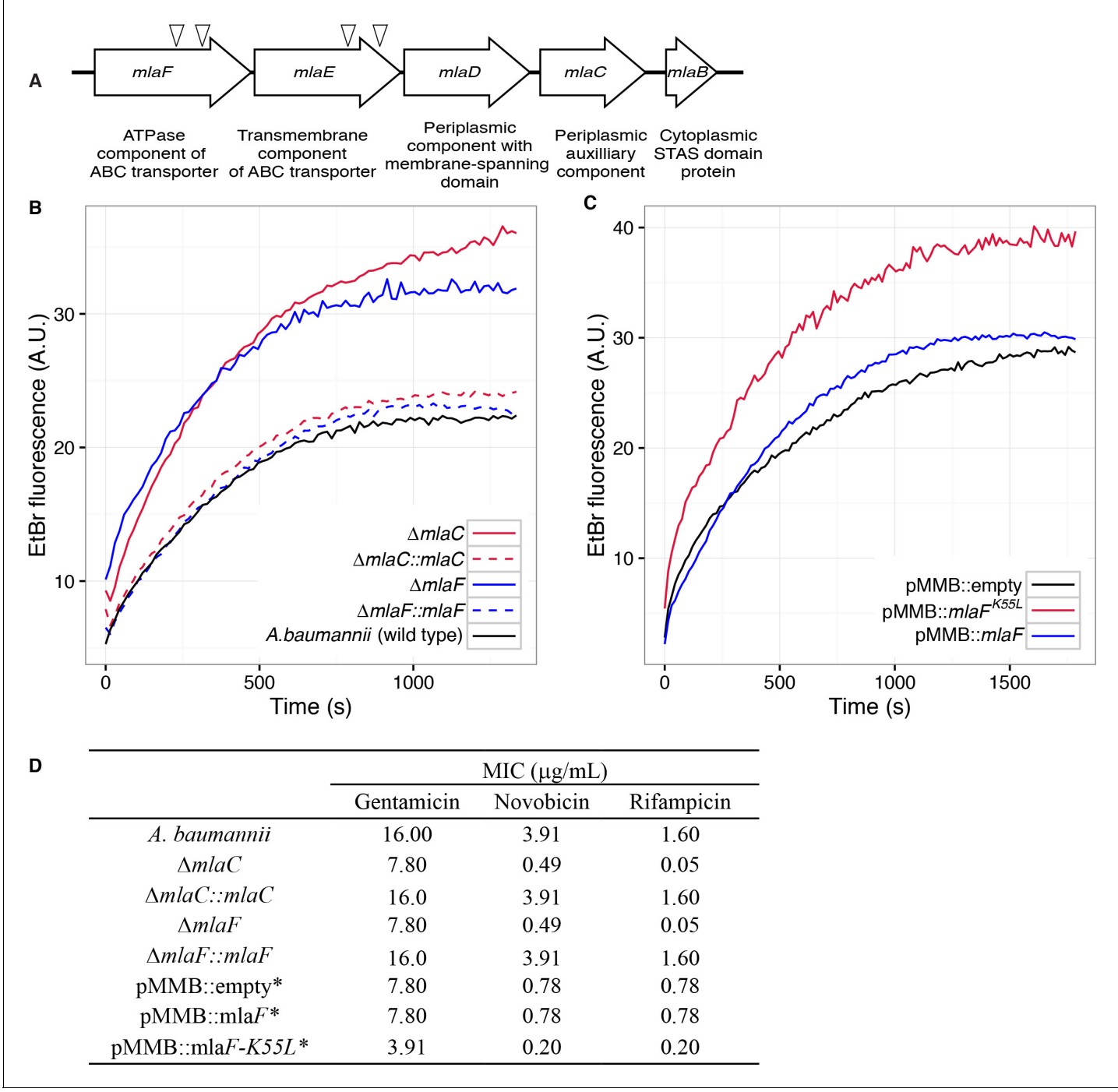

**Figure 1.** Disruption of the Mla system results in an altered outer membrane barrier. (**A**) Genomic organization of the *A. baumannii mlaFEDCB* operon and its predicted products. Triangles indicate the position of four independent transposon insertions, isolated in a screen for genes involved in outer membrane integrity. (**B**) Ethidium bromide uptake assay of outer membrane permeability of Δ*mla* mutants and complemented strains. A.U., arbitrary units. Lines shown depect the average of three technical replicates. (**C**) Ethidium bromide uptake assay of outer membrane permeability following plasmid-based expression of MlaF, compared to its dominant negative version, MlaF[K55L]. Lines shown depict the average of three technical replicates. (**D**) Minimum inhibitory concentration (MIC) of select antibiotics in *A. baumannii*. *Indicates wild type *A. baumannii* containing pMMB plasmid constructs, and cultures grown with the addition of kanamycin (25 µg/mL) to maintain plasmids and 50 µM IPTG for induction.

DOI: https://doi.org/10.7554/eLife.40171.003

The following figure supplements are available for figure 1:

**Figure supplement 1.** Disruption of the Mla system leads to an increase in exopolysaccharide production.

DOI: https://doi.org/10.7554/eLife.40171.004

*Figure 1 continued*

**Figure supplement 2.** Phase microscopy images of wild type and mla mutant *A. baumannii.*
DOI: https://doi.org/10.7554/eLife.40171.005

particularly essential for the hydrolysis of ATP. Mutations in this lysine residue are inhibited for nucleotide binding, and the mutated protein is rendered inactive (*Hanson and Whiteheart, 2005*). Additionally, ATPase mutants in the key lysine residue have been shown to have a dominant-negative effect on ATP hydrolysis when co-expressed with their wild-type ATPase counterparts, as typical ABC transporters have a structural requirement for two functional nucleotide-binding proteins which dimerize upon substrate transport (*Davidson and Sharma, 1997*; *Wilkens, 2015*).

Therefore, we created a version of the MlaF coding sequence with a leucine substitution of the Walker A lysine residue (MlaF$^{K55L}$), and then cloned the mutated *mlaF* into the low-copy pMMBkan vector under control of the *mlaF* native promoter. We observed that expression of MlaF$^{K55L}$ in wild type *A. baumannii* had a dominant-negative effect on membrane permeability as measured by EtBr uptake (*Figure 1C*), and expression of MlaF$^{K55L}$ also resulted in increased exopolysaccharide production as demonstrated by increased staining by crystal violet (*Figure 1—figure supplement 1B*). Correspondingly, expression of MlaF$^{K55L}$ rendered *A. baumannii* more sensitive to a variety of antibiotics, resulting in reduced MICs when compared to *A. baumannii* expressing the empty pMMBkan vector (*Figure 1D*). Therefore, expression of a defective ATPase results in a dominant-negative mutant with a comparable phenotype to deletion of components of the *mla* operon. These results demonstrate a requirement for ATP hydrolysis by MlaF for the maintenance of OM barrier function in *A. baumannii,* and indicate that the phenotypes of deletion mutants were likely a result of a lack of transport function, rather than formation of a toxic incomplete membrane protein complex.

## Structure of the *A. baumannii* MlaBDEF complex

The genetic arrangement and conservation of the components of this ATPase-containing transport complex indicated it was likely that the individual components formed a higher order protein structure. To define whether the Mla components form a stable protein complex, we expressed the entire operon (*mlaFEDCB*) from *A. baumannii* ATCC 17978 in *E. coli* with a carboxy-terminal hexahistidine tag on the MlaB component. Affinity purification of MlaB revealed three additional bands, with sizes corresponding to MlaF, MlaD, and MlaE (*Figure 2—figure supplement 1*) and confirmed by MALDI-TOF mass spectrometry analysis, indicating that these four proteins form a stable complex. We did not detect MlaC, suggesting it might interact only transiently with the other components, consistent with results recently reported by *Thong et al. (2016)*.

We next used cryo-electron microscopy to characterize the architecture of the *A. baumannii* MlaBDEF complex (abMlaBDEF). This complex is uniformly dispersed in vitreous ice (*Figure 2—figure supplement 2A*), and 2D classification demonstrated the presence of a range of views suitable for structure determination (*Figure 2—figure supplement 2B*). Following 2D- and 3D-classification, we obtained a final dataset of ~14,000 particles with which we obtained a structure to a resolution of 8.7 Å (*Figure 2—figure supplement 2D*). The structure possesses significant visible features in agreement with the nominal resolution (*Figure 2—figure supplement 2C*). Based on the bioinformatically-predicted localization of individual proteins and work recently performed on the similar *E. coli* Mla complex (ecMlaBDEF) (*Thong et al., 2016*), we propose that MlaD localizes to the periplasmic side of the IM, MlaE forms the central transmembrane region, and MlaF and MlaB form the bottom layer on the cytoplasmic face of the IM with two visible hetero-dimers (*Figure 2—figure supplement 2E*). We note that the structure of ecMlaBDEF, at lower resolution, was reported recently (*Ekiert et al., 2017*). The overall features of both structures, solved independently, are identical, suggesting that they correspond to the correct structure for the complex. However, the limited resolution of the ecMlaBDEF complex structure did not allow modeling of its individual subunits, in contrast to the abMlaBDEF structure reported here.

We note that a clear six-fold symmetry is present for the region of the map attributed to MlaD (*Figure 2B*), despite the fact that we only imposed a 2-fold symmetry averaging. This agrees with the proposed hexameric state of its *E. coli* homologue (ecMlaD) (*Thong et al., 2016*). We next modeled abMlaD, using an evolution restraints-derived structural model of ecMlaD (*Ovchinnikov et al.,*

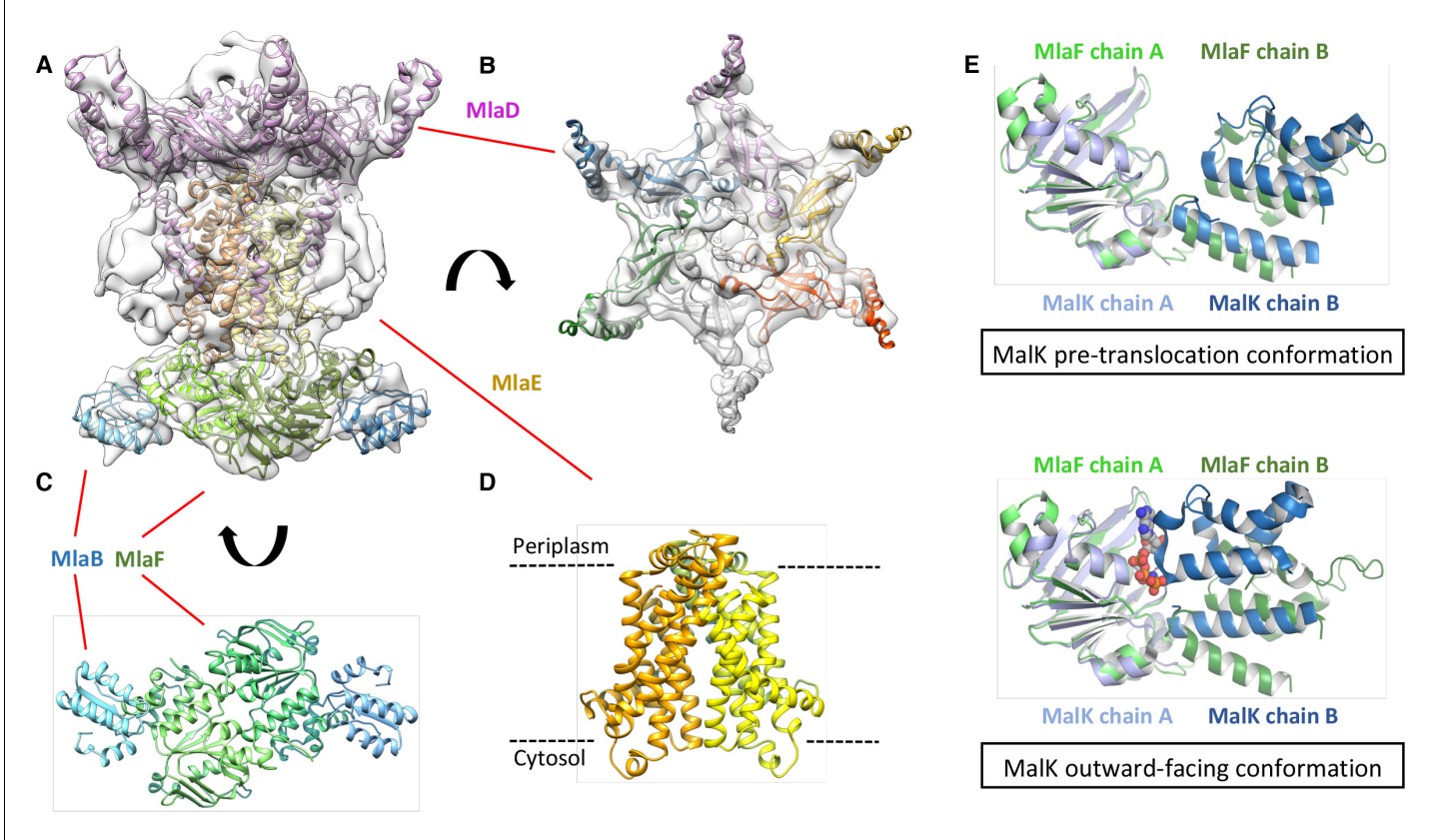

**Figure 2.** Structure of the abMlaBDEF complex. (**A**) Cryo-EM map of abMlaBDEF (grey), with structural models for MlaD, MlaB and MlaF (in magenta, cyan and green respectively) docked at their putative location. The density for the TM helices is clearly resolved. (**B–D**) Cartoon representation of the MlaD hexamer (**B**), the MlaB-MlaF hetero-tetramer (**C**), and the MlaE dimer (**D**) region of the abMlaBDEF atomic model. (**E**) Comparison of the MlaF domain arrangement in the EM map to that of the Maltose transporter ATPase MalK. The two chains of MlaD (in light and dark green) superimpose well to those of MalK (in cyan and dark blue) in the pre- translocation conformation (top, PDB ID: 4 KHZ), while a clear rotation is observed compared to the ATP-bound outward-facing conformation (bottom, PDB ID: 4KI0).
DOI: https://doi.org/10.7554/eLife.40171.006

The following figure supplements are available for figure 2:

**Figure supplement 1.** Mla components copurify following protien expression.
DOI: https://doi.org/10.7554/eLife.40171.007
**Figure supplement 2.** Cryo-EM structure of the abMlaBDEF complex.
DOI: https://doi.org/10.7554/eLife.40171.008
**Figure supplement 3.** Modeling of the abMlaD hexamer.
DOI: https://doi.org/10.7554/eLife.40171.009
**Figure supplement 4.** Close-up view of the MlaB, MlaD, MlaE and MlaF models in the abMlaBDEF cryo-EM map.
DOI: https://doi.org/10.7554/eLife.40171.010

*2017*) as a template, and used our previously-published EM-guided symmetry modeling procedure (*Bergeron et al., 2013*) to model its hexameric state. The obtained abMlaD hexameric model is at a low-energy minimum (*Figure 2—figure supplement 3B*) and fits the EM map density well (*Figure 2B* and *Figure 2—figure supplement 4B*). A crystal structure of the periplasmic domain of ecMlaD published recently (*Ekiert et al., 2017*) formed a crystallographic hexamer, suggesting that this corresponds to the native hexomeric arrangement for this domain. Our abMlaD hexameric model is very similar to the crystallographic ecMlaD structure (*Figure 2—figure supplement 3C*), supporting the proposed domain arrangement in the MlaBDEF complex. We note, however, that one region of density in the EM map is not accounted for by our MlaD hexamer model (*Figure 2B*). The localization of this extra density suggests that it corresponds to a ~ 45 amino-acid insert present

between strands 4 and 5 of the abMlaD β-sheet (*Figure 2—figure supplement 4A*). The role of this insert, uniquely found in the *A. baumannii* orthologue, is not known.

We next modeled the structures of MlaB and MlaF and fitted their respective coordinates in the corresponding region of the EM map (*Figure 2C* and *Figure 2—figure supplement 3A*). For both proteins, most helices are well resolved, which allowed us to place the models unambiguously. We then compared the conformation of the ATPase MlaF to that of the maltose transporter ATPase MalK, which has been trapped in several conformations of the transporter; that is the inward-facing state, the pre-translocation state, and the outward-facing state (*Khare et al., 2009*; *Oldham et al., 2013*). Interestingly, the arrangement of MlaF clearly resembles the pre-translocation state of MalK (*Figure 2D*). This suggests that we have trapped a similar conformation of the abMlaBDEF complex. It is possible that MlaD and/or MlaF, for which there are no equivalent in other ABC transporters, stabilizes this conformation. Alternatively, it is possible that the presence of detergents, which were present to solubilize the complex, mimics the natural ligand in the transporter's active site. Finally, the transmembrane (TM) region of the map is well resolved, and density for the transmembrane (TM) helices can be clearly identified. We therefore modeled abMlaE, using an evolution restraints-derived structural model of ecMlaE (*Ovchinnikov et al., 2017*) as a template, and fitted the obtained coordinates in the corresponding region of the map, with the orientation corresponding to the predicted topology. The resulting MlaE dimer model (*Figure 2D*) fits well to the EM map density (*Figure 2—figure supplement 4C*), and clearly corresponds to a closed transporter, with no solvent channel between the subunits. Interestingly, we also noted clear density for three TM helices that likely correspond to the MlaD N-terminal helices (*Figure 3A*). However, they lacked continuity, and

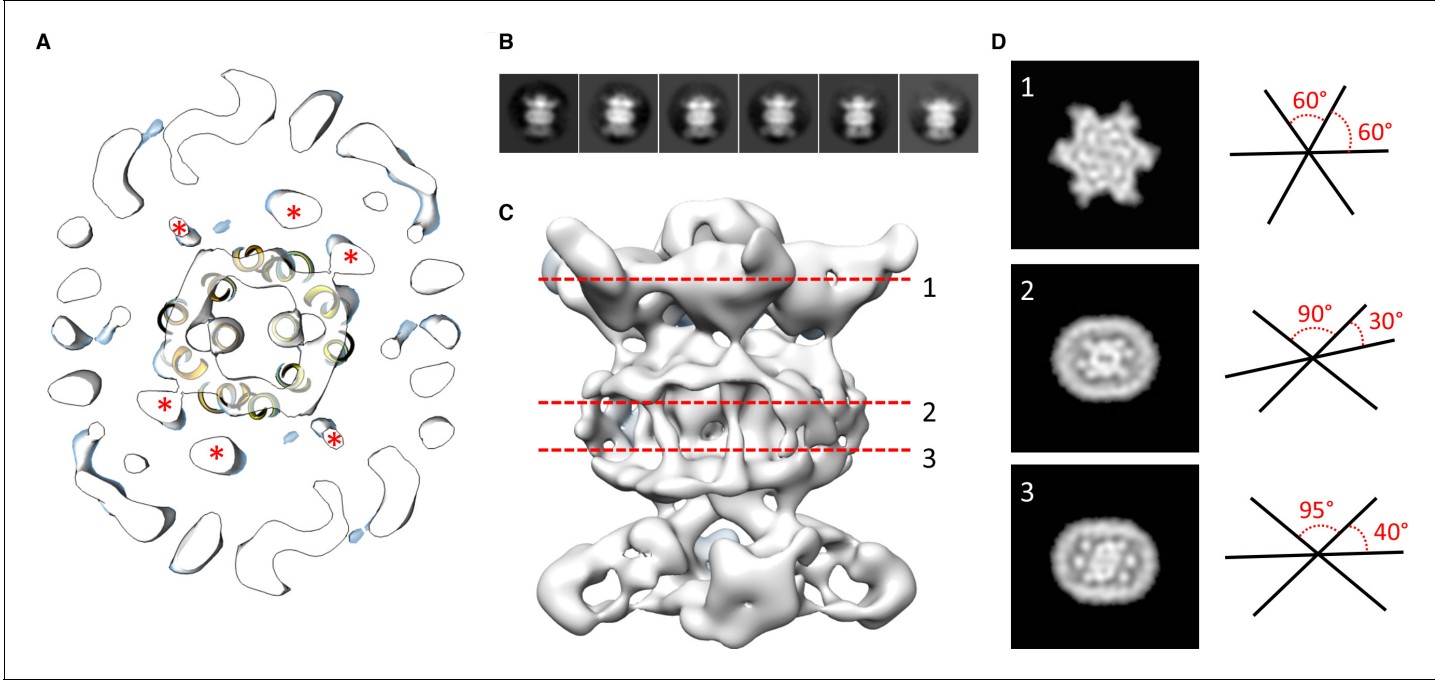

**Figure 3.** Localization of the 6 TM helices from MlaD. (**A**) lateral section of the abMlaBDEF EM map, with the MlaE model in yellow. Density attributed to the MlaD N-terminal helices are indicated with a red star. (**B**) 2D classes generated from the set of particles used to generate the abMlaBDEF structure, corresponding to side views. A range of orientations for the periplasmic domain is observed. (**C**) Structure of abMlaBDEF, generated using a subset of the most homogeneous ~8000 particles. Some features of the map shown in *Figure 3C* are not present, but the overall structure is similar. Six well-defined helices in the central TM region are visible. (**D**) Sections along the vertical axis, corresponding to the three red lines shown in C, is shown on the left. The six-fold axis of MlaD is visible in the periplasmic region, but this breaks down in the TM region, where the six helices are asymmetric. An angular representation of the six helices at each three cross-sections is represented on the right.

DOI: https://doi.org/10.7554/eLife.40171.011

The following figure supplement is available for figure 3:

**Figure supplement 1.** Comparison of the abMlaBDEF and ecMalBDEF complex.
DOI: https://doi.org/10.7554/eLife.40171.012

we observed that only two form a direct interaction with MlaE. It is possible that this is due to heterogeneity in the orientation of MlaD relative to the rest of the complex. To verify this, we performed further 2D classification of the particles used for reconstruction (*Figure 3B*), which revealed a range of positions for the MlaD region relative to the rest of the complex. We therefore performed further 3D classification leading to a smaller dataset of ~8000 particles. This produced a structure of lower resolution (~11.5 Å) but with the six MlaD N-terminal TM helices clearly visible (*Figure 3B*). While the periplasmic domain possesses 6-fold symmetry, the TM domains of MlaD do not appear symmetrical, with two forming close contacts with the density attributed to MlaE while the other four do not appear to contact any other proteins. This observation likely explains the asymmetry of contacts between the dimeric MlaE and the hexameric MlaD. A higher-resolution structure will be required to determine if additional contacts are formed between the outward-facing loops of MlaE and the periplasmic domain of MlaD.

## Components of the mla system interact directly with GPL

The crystal structure of MlaC has been solved from *Ralstonia solanacearum*. The structure contains a single phosphatidylethanolamine molecule oriented such that the hydrophobic acyl chains are located inside the protein while the hydrophilic head group is exposed (*Huang et al., 2016*). More recently, the crystal structure for MlaC has been solved from *E. coli* and shown to bind lipid tails (*Ekiert et al., 2017*). As noted in previous work performed on the *E. coli* Mla system, this is strong evidence that the substrates of the Mla system are GPL (*Malinverni and Silhavy, 2009*). In order to confirm that the periplasmic components of the Mla pathway in *A.baumannii* interact with GPL, we purified the soluble domains of both MlaC and MlaD by expressing histidine-tagged proteins followed by Ni-affinity FPLC purification. After overnight dialysis of the proteins, we performed Blighdyer chloroform extraction on the purified proteins to isolate any bound GPL and analyzed the results by LC-MS/MS. GPL analysis revealed both phosphatidylglycerol and phosphatidylethanolamine of varying acyl chain lengths. This suggests the possibility that the periplasmic substrate binding components of the system may bind diacylated GPL molecules with limited polar head group specificity (*Figure 4—figure supplement 1*).

## Mla mutants have decreased abundance of outer membrane GPL

Given the OM defect of *mla* mutants, as well as the system's apparent direct association with GPL, we chose to further characterize the overall membrane GPL composition of the *mla* mutants. Previous work on the Mla system in *E.coli* has demonstrated an increase in hepta-acylated lipid A in *mla* mutants, indicating activation of PagP that acylates GPL and lipid A in the outer leaflet of the OM in enterobacteria (*Malinverni and Silhavy, 2009*; *Dalebroux et al., 2014*). From this data it has been suggested that the system may serve to maintain lipid asymmetry within the OM, although it is well known that GPL displacement to the OM outer leaflet is a general reflection of chemical damage to the OM (*Jia et al., 2004*; *Bishop et al., 2000*; *Dekker, 2000*). However, biochemical analysis of the membrane GPL composition for *mla* mutants has not been published for any organism to our knowledge, so we sought to apply our lab's methods of GPL quantification to test the hypothesis of retrograde transport function. To determine whether *A. baumannii mla* mutations cause changes in the membrane GPL concentration, GPL were extracted from inner and outer membrane fractions separated by density centrifugation. As can be seen on *Figure 5—figure supplement 3*, density centrifugation results in nice separation of the outer and inner membranes of *Acinetobacter baumannii*, with the OM contain the vast majority of OmpA and the inner membrane containing all the NAPPH oxidase. Thin-layer chromatography (TLC) and electrospray-ionization time-of-flight mass spectrometry (ESI-MS) were used to qualitatively assess GPL composition from these well separated membrane fractions. TLC and ESI-MS indicated Δ*mlaC A. baumannii* had a dramatically decreased abundance of all major phospholipid species in the OM compared to wild type. (*Figure 4A* and *Figure 4—figure supplement 2*).

To better analyze the differences in membrane GPL, we quantified GPL by normal phase liquidchromatography collision-induced-dissociation mass spectrometry (LC-MS/MS). We quantified the ratio of individual GPL within each membrane by normalizing to an internal standard of known quantity. We then normalized the quantified GPL to the protein content of isolated IM and OM. Quantitative LC-MS/MS confirmed the overall reduction in outer membrane GPLs observed by ESI-MS and

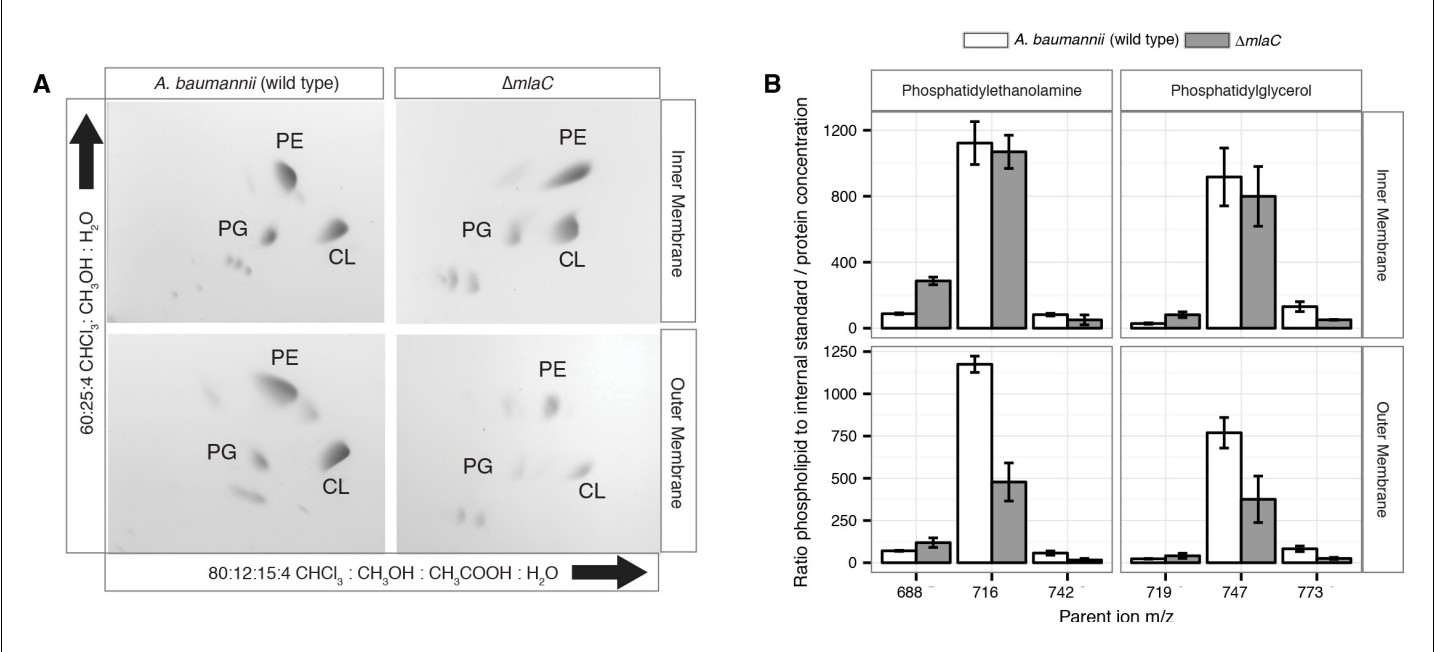

**Figure 4.** Outer membrane glycerophospholipid levels are reduced in ΔmlaC mutant. (**A**) Identification of inner and outer membrane phospholipids of wild type *A. baumannii* and ΔmlaC using 2D thin-layer chromatography. PE, phosphatidylethanolamine; PG, phosphatidylglycerol; CL, cardiolipin. (**B**) LC-MS/MS quantification of isolated inner and outer membrane glycerophospholipids. Error bars indicate ±s.e.m. (n = 3).

DOI: https://doi.org/10.7554/eLife.40171.013

The following figure supplements are available for figure 4:

**Figure supplement 1.** Purified periplasmic components of the Mla system remain bound to glycerophospholipids.

DOI: https://doi.org/10.7554/eLife.40171.014

**Figure supplement 2.** Deletion of mlaC results in a reduction in levels of outer membrane glycerophospholipids.

DOI: https://doi.org/10.7554/eLife.40171.015

TLC, with the reduced levels observable across multiple GPL species for ΔmlaC mutants relative to wild type (*Figure 4B*). Therefore, mutations in the components of the Mla system result in a decrease in OM GPL, whereas the retrograde transport hypothesis would predict an increase in OM GPL. Therefore, these results instead suggest a possible role for Mla in outward GPL trafficking.

## Mla mutants demonstrate an accumulation of newly synthesized GPL in the IM

The overall decrease in outer membrane glycerophospholipids of A. baumannii *mla* mutants suggests that either the Mla system is functioning to deliver GPLs from the inner membrane to the outer membrane, or alternatively, mutations in the Mla system may disrupt the outer membrane in a manner that leads to the activation of outer membrane phospholipases, which then degrade GPL. Work performed on the Mla system in *E.coli* has demonstrated that disruption of genes in the Mla pathway results in activation of both the OM acyl-transferase PagP, which cleaves a palmitate moiety from GPL and transfers it to LPS and PG, creating a hepta-acylated LPS molecule and palmitoyl-PG and the OM phospholipase PldA (*Malinverni and Silhavy, 2009*; *Bishop et al., 2000*). *A. baumannii* has no known PagP enzyme but similar activity of the multiple predicted OM phospholipases could account for the reduction in OM GPL as observed by TLC and quantitative mass spectrometry. Therefore, we designed a mass spectrometry-based assay to study intermembrane GPL transport using $^{13}$C stable isotope labeling (*Figure 5—figure supplement 1A*), to better analyze the directionality of GPL transport by the Mla system between the bacterial membranes. When grown in culture with sodium acetate as the sole carbon source, many bacteria directly synthesize acetyl-CoA using the conserved enzyme acetyl-CoA synthase (*Kumari et al., 2000*). Acetyl CoA, the precursor metabolite for fatty acid biosynthesis, is first converted to malonyl-CoA and enters the FasII (fatty acid biosynthesis) pathway that supplies endogenously synthesized fatty acids to macromolecules such as

lipopolysaccharides, phospholipids, lipoproteins, and lipid-containing metabolites. By growing cultures in unlabeled acetate then 'pulsing' with 2-$^{13}$C acetate and analyzing separated membrane fractions from set time points, we can observe the flow of newly synthesized GPLs between the IM and OM of *A. baumannii* (*Figure 5—figure supplement 1B*) (*Dalebroux et al., 2014*).

Upon introducing the 2-$^{13}$C acetate as the sole carbon source, $^{13}$C-labeled GPL were immediately synthesized in the bacterial cytoplasm. We reasoned that continued growth in $^{13}$C acetate should result in a mixed pool of unlabeled and labeled IM GPL molecules. As the GPL are then fluxed from the IM to the OM, the likelihood that an individual GPL molecule is transported is directly proportional to the ratio of labeled to unlabeled GPL in the IM pool. As the bacteria continue to grow in $^{13}$C acetate, the ratio of labeled to unlabeled GPL in the IM will gradually increase as new GPL are synthesized and inserted in the IM. As such, the likelihood of transporting labeled GPL to the OM will also increase. A comparison of the ratios of labeled to unlabeled GPL in the IM and OM will thus reflect the efficiency of transport between the membranes, and analysis of transport in wild type *A. baumannii* will establish reference for transport efficiency with which to compare our mutants. Additionally, OM phospholipases, some of which may be activated upon membrane damage (*Istivan and Coloe, 2006*), will not distinguish between labeled and unlabeled GPL and therefore will not affect the ratio of labeled to unlabeled GPL obtained from this assay.

Membrane separation and analysis of wild type *A. baumannii* revealed near-identical rates-of-change between the two membranes in ratios of $^{13}$C-labeled to unlabeled GPLs, indicating that newly synthesized GPLs are transported and inserted into the OM at a rate equivalent to their rate of synthesis and assembly within the IM. Furthermore, the ratios of labeled to unlabeled GPLs were nearly equal in the IM compared to the OM at the time points evaluated, indicating that GPL transport likely occurs rapidly, consistent with earlier pulse-chase experiments performed in *E. coli* that estimate the half-life of translocation of various GPLs at between 0.5 and 2.8 min (*Donohue-Rolfe and Schaechter, 1980*). By contrast, mutants in the Mla system accumulate newly synthesized GPLs in their IM at a greater rate than occurs in the OM as evidenced by the increasing disparity in the ratio of labeled to unlabeled GPLs between the IM and OM over time (*Figure 5A*). The discrepancy in ratios of labeled to unlabeled GPLs between the IM and OM of Δ*mlaF* is apparent for PG and PE of varying acyl chain lengths corresponding to the most naturally abundant species C16:0/C16:0, C18:1/C18:1, or C16:0/C18:1 (*Supplementary file 2*). Further, the effects of MlaF$^{K55L}$ expression on GPL trafficking were similar to what was observed in the Δ*mlaF* strain (*Figure 5B*). Therefore, ATP hydrolysis by MlaF appears to be a requirement for extraction of these GPLs from the IM of *A. baumannii* for subsequent transport to the OM.

To better characterize the role of the periplasmic substrate binding component MlaC, we performed similar stable isotope pulse experiments to observe the flow of newly synthesized GPLs in the Δ*mlaC* strains. Stable isotope experiments on Δ*mlaC* mutants reveal IM accumulation of newly synthesized GPLs similar to the result in Δ*mlaF* mutants (*Figure 5—figure supplement 2A*), indicating that in the absence of the periplasmic component GPLs are not efficiently removed from the IM by the remainder of the Mla system. We also sought to characterize the potential role of the putative OM-lipoprotein MlaA, which has been implicated as a component of the Mla system in *E. coli*. A chromosomal deletion strain of *mlaA* was created by allelic exchange, and complemented by expression of MlaA from a pMMB67EH-Kan plasmid. The results of the stable isotope pulse experiments in the Δ*mlaA* strain revealed results consistent with those obtained from Δ*mlaC* and Δ*mlaF*, in which the ratio of labeled to unlabeled GPL is consistently higher in the inner membrane than the outer membrane after one hour of exposure to $^{13}$C-acetate (*Figure 5—figure supplement 2B and C*). These results are consistent with a model in which the IM-localized ABC transporter complex MlaB-DEF first transfers GPLs to the periplasmic binding protein MlaC, which then transports GPL to the OM, whereupon MlaA facilitates GPL insertion into the OM.

## Discussion

We performed a screen to identify *A. baumannii* proteins that are essential for its OM barrier that led to the identification of an ABC transport system whose ATPase activity maintains OM barrier function. IM and periplasmic components of this system can be purified, bind GPLs, and assemble into a defined protein complex with significant symmetry, indicating that this system could function to transport GPLs from the IM to the OM. Consistent with the possibility that Mla functions as an

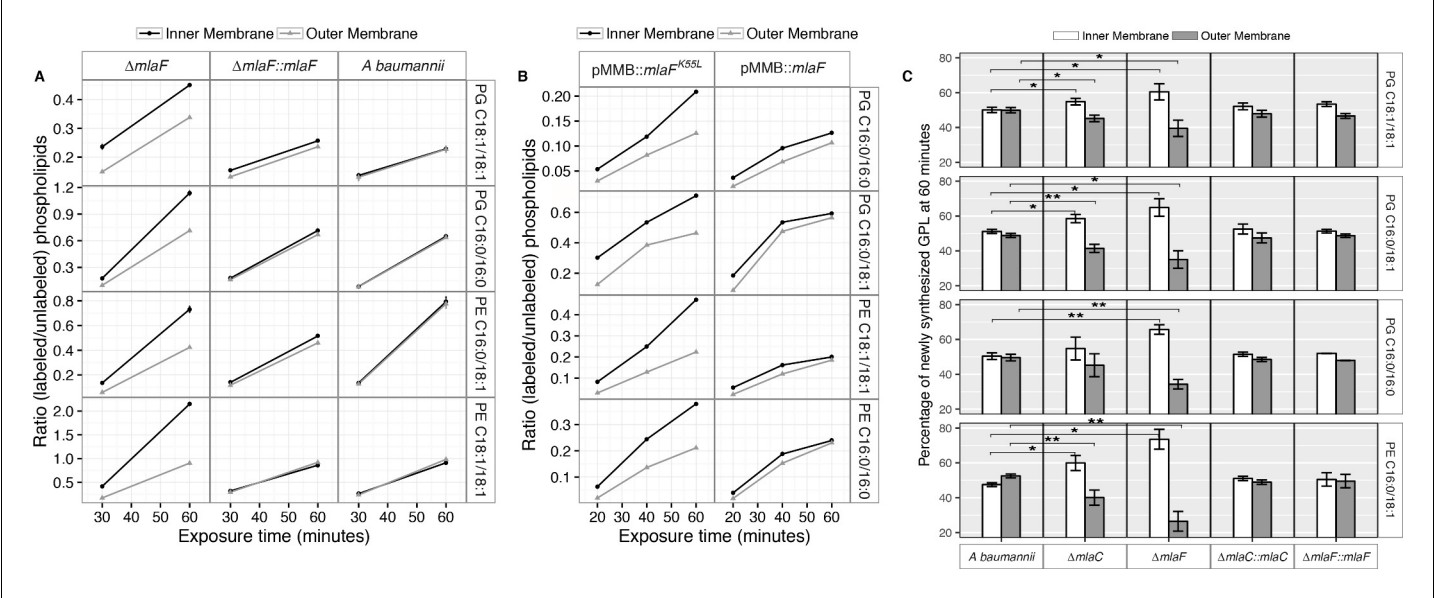

**Figure 5.** Newly synthesized glycerophospholipids accumulate at the inner membrane of Mla mutants. (**A**) LC-MS/MS quantification of 13C labelled/unlabeled glycerophospholipids in isolated membrane fractions over time after growth in 2–13C acetate in Δ*mlaF* and complemented strain. Facet labels on the right indicate the specific glycerophospholipid species analyzed and the acyl chain length. PG, phosphatidylglycerol; PE, phosphatidylethanolamine. Shown is representative data from repeated experiments. (**B**) LC-MS/MS quantification of 13C labelled/un- labeled glycerophospholipids in isolated membrane fractions following plasmid-based expression of MlaF compared to its dominant negative version, MlaF$^{K55L}$. Facet labels on the right indicate the specific glycerophospholipid species analyzed and the acyl chain length. PG, phosphatidylglycerol; PE, phosphatidylethanolamine. Shown is representative data from repeated experiments. (**C**) Relative proportion of newly synthesized GPL on IM and OM after one hour growth in 2–13C acetate. Error bars represent ± s.d. (n = 2). Statistical analyses performed using a Student's *t* test. *p*-Value: *, p < 0.05; **, p < 0.01.

DOI: https://doi.org/10.7554/eLife.40171.016

The following figure supplements are available for figure 5:

**Figure supplement 1.** A stable isotope assay of glycerophospholipid transport from the inner membrane to the outer membrane.
DOI: https://doi.org/10.7554/eLife.40171.017

**Figure supplement 2.** Newly synthesized glycerophospholipids accumulate at the inner membrane of MlaA mutants.
DOI: https://doi.org/10.7554/eLife.40171.018

**Figure supplement 3.** Confirmation of inner and outer membrane separation.
DOI: https://doi.org/10.7554/eLife.40171.019

anterograde transporter, the OM of mutants show an overall reduction of GPL along with an excess accumulation of newly synthesized GPL on the IM. Therefore, these results lead us to propose that the function of the *A. baumannii* Mla system is the trafficking of GPL from the IM, across the periplasm, for delivery to the outer membrane (*Figure 6*). According to this model, ATP hydrolysis by MlaF provides the initial energy to extract GPL from the IM, while the substrate binding components MlaD and MlaC take up lipids for delivery to the OM. It has been observed by van Meer and colleagues that complete extraction of GPLs from the membrane bilayer requires a Gibbs free energy of ~100 kJ/mol (*Abreu et al., 2004*; *van Meer et al., 2006*), whereas ATP contains just 30 kJ/mol of energy. To account for the energy difference, a hydrophobic acceptor molecule is proposed to allow the lipids to fully dissociate from the rest of the ABC transporter and facilitate complete removal from the bilayer. The GPL-binding component, MlaD, contains an IM spanning domain and is shown here, and in orthologous systems, to be in complex with the MlaE and MlaF proteins within the IM (*Ekiert et al., 2017*; *Roston et al., 2012*). The close association of MlaD with the outer leaflet of the IM may allow it to extract lipids from the IM by hydrophobic interaction with the acyl chains after ATP hydrolysis by MlaF. Subsequent trafficking across the periplasm then involves the periplasmic GPL binding protein MlaC, which likely accepts GPL from MlaD and then carries them to the OM. We note however the observed effect of *mlaC* deletion on GPL accumulation in the IM, while statistically significant for most of the analyzed diacyl-glycerophospholipids, appears to be less than that of

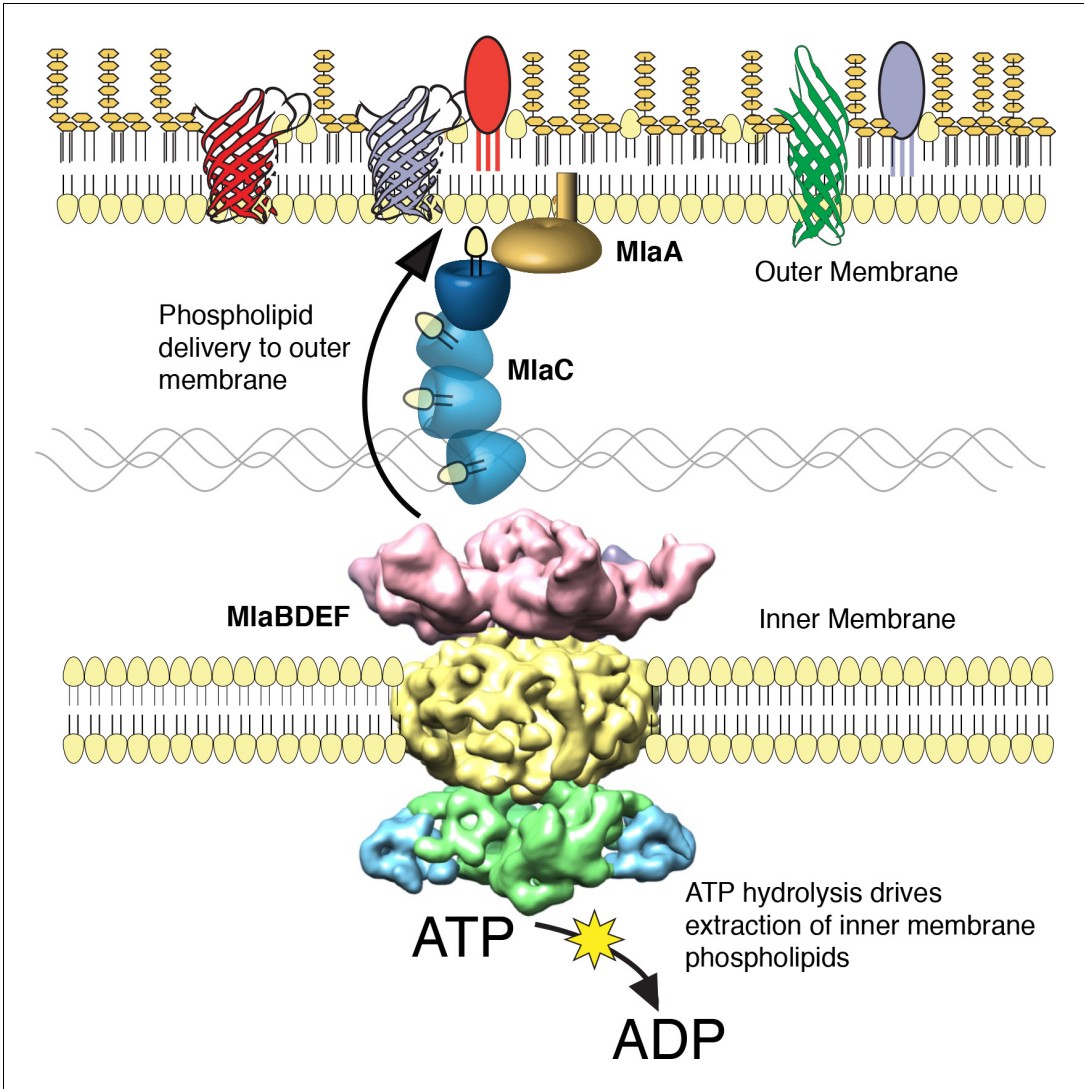

**Figure 6.** The multicomponent Mla system transports glycerophospholipids from the inner membrane to the outer membrane of *A. baumannii*. A schematic of glycerophospholipid transport to the Gram-negative bacterial outer membrane by the Mla system.
DOI: https://doi.org/10.7554/eLife.40171.020

deletion of the ATPase component (*Figure 5C*), suggesting that while MlaC may participate in transfer to the OM, there may be redundant mechanisms by which the IM complex can transport or remove IM GPL in the absence of MlaC. While the precise mechanism of GPL insertion into the OM is not yet known, work performed on the *E.coli* Mla system has shown that MlaC interacts with both the IM MlaFEDB complex, as well as with the putative OM lipoprotein MlaA, and our results support a role for MlaA in the function of the overall Mla system and delivery of GPL to the OM.

In this work, we designed a method to monitor lipid transport between Gram-negative bacterial membranes using stable $^{13}$C isotope labeling. We considered the possibility of loss of GPL from the outer membrane due to outer membrane vesicle formation or, more likely, by the possible increased activation of outer membrane phospholipases. Both would have the effect of removing GPL from the outer membrane and would result in lower GPL levels in the outer membrane of *mla* mutants. Neither of these mechanisms would operate specifically on either labeled or unlabeled lipids. We

understood the decrease in outer membrane GPL to be insufficient evidence of an anterograde transport function for *mla*, and for this reason we developed the stable isotope assay to control for these possibilities. The stable isotope assay gives insight into whether these results are due simply to mislocalization or degradation of outer membrane GPL, or if they can in fact be attributed to deficient anterograde GPL transport. This is because the peak intensity of each newly synthesized, $C^{13}$-labeled GPL is normalized to the corresponding unlabeled version of that GPL species with each sample injected into the LC-MS/MS. The result is a ratio of labeled to unlabeled GPL for every membrane sample. With $C^{13}$-acetate as the sole carbon source, we observe a gradual increase in the ratio of labeled GPL relative to unlabeled GPL over time. In wild type bacteria, these ratios for the inner and outer membranes track closely over time, which indicates that under these conditions GPL transport from the inner to the outer membrane occurs quite rapidly. In *mla* mutants the ratio is both higher and typically increases at a greater rate in the inner membrane. Phospholipases and budding outer membrane vesicles will not distinguish between labeled and unlabeled GPL species, and so will not impact the ratios obtained with this assay.

Our results using this assay are consistent with the Mla system functioning as an anterograde GPL transporter, however they do not exclude the possibility of a dual role for Mla components in the maintenance of OM lipid asymmetry. Previous work performed on the orthologous Mla system in *E. coli* has been interpreted to suggest that the function of the system is to remove GPL from the outer leaflet of the OM for retrograde transport back into the cytoplasm based on the observation that *E. coli mla* mutants likely contain GPLs on the outer leaflet of the OM. (*Malinverni and Silhavy, 2009*; *Benning, 2008*). This proposed function was inferred from the observation that gene deletions resulted in an increased activation of the OM-phospholipase enzymes PagP and OMPLA, suggesting an increased amount of GPL in the outer leaflet of the OM (*Malinverni and Silhavy, 2009*). The interpretation of retrograde transport function was also based on the existence of an orthologous system in plant chloroplasts that transports lipids from the endoplasmic reticulum (ER) into the organelle. Many plants require this retrograde transport function because certain lipids in the chloroplast thylakoid membrane derive from GPL originating in the ER (*Hurlock et al., 2014*). However, since Gram-negative bacteria synthesize GPL within the IM, retrograde transport of GPL would only be necessary for the recycling of GPL mislocalized to the OM outer leaflet. Although this is a reasonable inference based on data available at the time, we would point out that the directionality of transport by the *E. coli* Mla system had not been thoroughly probed experimentally using membrane analysis or with a functional assay of the type performed here. It is conceivable that the import function of the orthologous chloroplast TGD system is a result of adaptation to the intracellular environment, the system in this case having evolved to aid in the transfer of GPL from the nearby ER to the chloroplast. Furthermore, while it is possible that the Mla system in *E. coli* serves a different primary function than in *A. baumannii,* we demonstrate here that both complexes possess a similar architecture, pointing to a conserved function. The outer membrane defect phenotypes observed in *E. coli mla* mutants might also be explained by a disruption of OM structure stemming from decreased concentrations of OM GPL, leading to activation of the PagP enzyme. It is well established that for *E. coli,* GPL displacement to the OM outer leaflet and subsequent activation of these enzymes reflects OM instability and can be achieved by chemical disruption of the bilayer (*Jia et al., 2004*; *Bishop et al., 2000*; *Dekker, 2000*). It may be the case that the OM of *E. coli mla* mutants contain GPL in the outer leaflet, but the possibility remains that OM GPL can flip into the outer leaflet under conditions of OM damage resulting from an imbalance of LPS-to-GPL ratios, along with perhaps the corresponding disruption of OM proteins. However, final determination of the directionality of GPL transport by the Mla system in *E.coli* and other organisms will require intermembrane transport studies similar to what has been done here for *A. baumannii,* along with studies similar to those performed for the Lpt LPS transport system for which molecular transfer of LPS from molecule to molecule of the Lpt system is functionally defined.

Following the introduction of the retrograde transport model for Mla function into the existing literature, a number of studies have examined the phenotypic effects of Mla disruption in various organisms. In a recent study in PNAS, Powers and Trent first obtained *A. baumannii* deficient in lipooligosaccharide (LOS) by selection in the presence of polymyxin B (*Powers and Trent, 2018*). They then performed an evolution experiment, passaging the strains in cultures containing polymyxin B over 120 generations, at which point they observed significantly improved growth in the populations. These evolved populations were also observed to have increased resistance to antibiotics

including vancomycin, bacitracin, and daptomycin, and to appear more morphologically consistent relative to the unevolved strains when observed microscopically. Whole genome sequencing of the evolved strains revealed mutations in *mla* genes in seven of the 10 evolved populations. They also observed frequent disruptions in *pldA*, as well as in other envelope genes. To further study these effects, they then introduced clean deletions of *mlaE* and *pldA* to ATCC 19606, and selected for LOS-deficient bacteria by plating on polymyxin B. These double mutants demonstrated improved growth and resistance to antibiotics but continued to display altered cellular morphology. The authors present their data as evidence in support of Mla as a retrograde transport system, and assume that a lack of removal of GPL from the outer leaflet is promoting the OM barrier. We would point out that lacking in their data is examination of the membrane glycerophospholipid (GPL) profile in their LOS-deficient mutants. It is assumed by the authors that *mla* and *pldA* mutations have the effect of stabilizing a symmetric outer membrane produced in the absence of LOS by allowing GPL to fill in the outer leaflet, resulting in improved growth and antibiotic resistance. Given that the data suggests that Mla and PldA are selected against when LOS is absent, examination of the outer membrane GPL content might have supported the authors' conclusions if it revealed an increase in GPL in *mla* and *pldA* mutants. Absent such data, it is not obvious to us that the authors have sufficiently ruled out alternative explanations for their observed phenomena. For example, we would question the mechanisms regulating the homeostasis of both the inner and outer membranes and the entire periplasmic space in the absence of LOS. The authors acknowledge earlier work that observed an increase in expression of *mla* genes upon initial loss of LOS in 19606 (*Henry et al., 2012*; *Boll et al., 2016*). Genes in the *mla* pathway were shown to have an up to 7.5-fold increase in gene expression upon loss of LOS. Powers and Trent assert that the function of Mla is deleterious in the absence of LOS, but perhaps what is deleterious is the profound upregulation of *mla* expression in the absence of LOS, combined with an active PldA. If Mla is an anterograde transporter, we can imagine this might create a situation in which GPL are rapidly removed from the inner membrane and then degraded in the outer membrane in excess of what the cell can support and limiting both of these processes together simply allows the cell to achieve a new homeostasis. Understanding of the myriad processes regulating bacterial outer membrane assembly and integrity remains limited even when LOS is present, and so interpreting results such as these as providing direct evidence of function may exceed the limits of the data.

The gene for MlaA, the proposed OM component, is at a different chromosomal location from the remainder of the *mla* operon. Recent structural studies on MlaA have revealed that MlaA forms a ring-shaped structure localized the inner leaflet of the OM, and have shown it to form stable complexes with the outer membrane proteins OmpF and OmpC (*Abellón-Ruiz et al., 2017*). The proposed structure of MlaA in the OM supports the argument that MlaA is involved in removal of GPL from the outer leaflet, and it is suggested that GPL from the outer leaflet travel through a pore in MlaA where they are received by MlaC, yet our data reveals that *A. baumannii ΔmlaA* mutants are defective in delivery of GPL from the IM to the OM. These data can be reconciled by a model in which MlaA functions both to remove mislocalized GPL from the outer leaflet of the OM, and additionally serves to facilitate delivery of GPL to the OM by MlaC, perhaps by enabling MlaC localization to the surface of the inner leaflet. By this model, mutations in MlaA will be phenotypically similar to mutations in other components of the Mla system, and we would expect to observe a decreased rate of anterograde GPL transport. We would here point out that while previous work has implicated the Mla system in the maintenance of OM lipid asymmetry through observation of increased activity of PagP, the role of the MlaFEDB complex and MlaC in retrograde GPL transport has previously only been inferred from homology to the chloroplast TGD system. It is established that cellular mechanisms exist in Gram-negative bacteria to resist stressful conditions that lead to OM disruption. For example, OM phospholipase enzymes, such as PldA, are activated under conditions of membrane stress to digest GPL in the outer leaflet of the OM, as high levels of GPL in the outer leaflet destabilize the OM barrier function. The model of retrograde GPL transport by the Mla system proposes that growing cells expend cellular energy in the form of ATP in order to transport undigested GPL from the OM, across the periplasm, and back into the IM, at which point some of those same molecules will be transported back to the OM by an unknown mechanism. However, the available data points most clearly to a model of anterograde GPL transport by MlaFEDB and MlaC, facilitated in some way by MlaA.

The first three genes of the *mla* operon – comprising an ATPase, permease, and substrate-binding components of the ABC transporter complex – are conserved in *Mycobacteria spp, Actinobacteria*, and chloroplasts, while the entire five-gene operon appears to be conserved in Gram-negative bacteria (*Casali and Riley, 2007*). Given the conservation of the system across Gram-negative species, our results may shed light on a generalized mechanism contributing to OM biogenesis. Additionally, we have here demonstrated that the function of this ABC transport system is crucial for maintaining the integrity of the *A. baumannii* OM. The fact that *mla* mutations are tolerated, and that levels of OM GPL are reduced but not abolished, suggests the intriguing possibility of additional undiscovered mechanisms of GPL delivery to the OM. Also of interest is the potential role of the increased exopolysaccharide observed upon disruption of the Mla system. It is possible this exopolysaccharide plays a partially compensatory role in *A. baumannii* resulting from decreased OM GPL, given that recent work has shown that *A. baumannii* exopolysaccharides can contribute to antibiotic resistance, likely through improved barrier function (*Geisinger and Isberg, 2015*).

The progression towards a more complete understanding of intermembrane GPL transport and OM barrier function should ultimately have relevance in the development of novel drug targets to undermine emerging antibiotic resistance in Gram-negative pathogens. The emergence of antibiotic resistant Gram-negative bacteria for which few or no antibiotics are available therapeutically is an important medical concern. This issue is typified by current isolates of *A. baumannii* that can only be treated with relatively toxic colistin antibiotics. This has led many individuals and agencies to propose the development of single agent antimicrobials which could be used for organisms such as *A. baumannii* and *P. aeruginosa* that have significant antibiotic resistance. Therefore, work furthering the understanding of the OM barrier could lead to the development of drugs which target the barrier and allow the therapeutic use of many current antibiotics.

# Materials and methods

**Key resources table**

| Reagent type (species) or resource | Designation | Source or reference | Identifiers | Additional information |
|---|---|---|---|---|
| Gene (*Acinetobacter baumannii*) | *mlaC* | NA | Genbank Accession: AKA30172.1 | |
| Gene (*A. baumannii*) | *mlaF* | NA | Genbank Accession: AKA30169.1 | |
| Gene (*A. baumannii*) | *mlaE* | NA | Genbank Accession: AKA30170.1 | |
| Gene (*A. baumannii*) | *mlaD* | NA | Genbank Accession: AKA30171.1 | |
| Gene (*A. baumannii*) | *mlaB* | NA | Genbank Accession: AKA30173.1 | |
| Gene (*A. baumannii*) | *mlaA* | NA | Genbank Accession: AKA32955.1 | |
| Gene (*A. baumannii*) | *phoU* | NA | Genbank Accession: AKA33305.1 | |
| Strain, strain background (*A. baumannii*) | *Acinetobacter baumannii* ATCC 17978 | *Pelletier et al., 2013* *Baumann et al., 1968* Source: ATCC | GenBank ACCESSION: CP000521 | |

*Continued on next page*

*Continued*

| Reagent type (species) or resource | Designation | Source or reference | Identifiers | Additional information |
|---|---|---|---|---|
| Genetic reagent (*A. baumannii*) | ATCC 17978 ΔphoU | This paper | | Chromosomal deletion in ATCC 17978 by allelic exchange using pEX2tetRA vector |
| Genetic reagent (*A. baumannii*) | ΔmlaF | This paper | | Chromosomal deletion in ATCC 17978 by allelic exchange using pEX2tetRA vector |
| Genetic reagent (*A. baumannii*) | ΔmlaC | This paper | | Chromosomal deletion in ATCC 17978 by allelic exchange using pEX2tetRA vector |
| Genetic reagent (*A. baumannii*) | ΔmlaA | This paper | | Chromosomal deletion in ATCC 17978 by allelic exchange using pEX2tetRA vector |
| Antibody | anti-OmpA (rabbit polyclonal) | This paper | | Made to purified OmpA by GenScript Biotech Corp (1:1000) |
| Recombinant DNA reagent | pMarKT (plasmid) | This paper | | Progenitors: C9 Himar (PCR), TetRA from Tn10 (PCR), pACYC184, pBT20. |
| Recombinant DNA reagent | pEX2tetRA (plasmid) | This paper | | Progenitors: tetRA (PCR), pEXG2 |
| Recombinant DNA reagent | pMMBkan (plasmid) | This paper | | Kanamycin resistance marker inserted at DraI site of pMMB67EH |
| Recombinant DNA reagent | pMMBkan: MlaF (plasmid) | This paper | | pMMBKan expressing *mlaF* |
| Recombinant DNA reagent | pMMBkan: MlaF$^{K55L}$ (plasmid) | This paper | | pMMBKan expressing Walker box mutant of *mlaF* |
| Recombinant DNA reagent | pET28a: MlaFEDCB-His (plasmid) | This paper | | pET28a expressing MlaFEDCB with C-terminal HISX6 tag on MlaB. |
| Recombinant DNA reagent | pET15b-mlaC-SD-His (plasmid) | This paper | | pET15b expression vector containing MlaC soluble domain with C-terminal HisX6 tag. |
| Recombinant DNA reagent | pET15b-mlaD-SD-His (plasmid) | This paper | | pET15b expression vector containing MlaD soluble domain with C-terminal HisX6 tag. |
| Chemical compound, drug | 2–13C acetate | Cambridge Isotope Laboratories, Inc. | | |
| Chemical compound, drug | BCIP-Toluidine (XP) | Gold Biotechnology | B-500–10 | |
| Chemical compound, drug | N-Phenyl-1-naphthylamine (NPN) | Sigma Aldrich | 104043–500G | |

*Continued on next page*

*Continued*

| Reagent type (species) or resource | Designation | Source or reference | Identifiers | Additional information |
|---|---|---|---|---|
| Chemical compound, drug | NADH | Sigma Aldrich | 606-68-8 | |
| Chemical compound, drug | CCCP (Carbonyl cyanide 3-chlorophenylhy drazone) | Sigma-Aldrich | C2759-1G | |
| Software, algorithm | MotionCorr2 | *Thong et al., 2016* | | Dr. Agard Lab, University of CA San Francisco |
| Software, algorithm | CTFFIND4 | *Rohou and Grigorieff, 2015* | | Dr. Grigorieff Lab, University of MA Medical Center |
| Software, algorithm | Appion | *Lander et al., 2009* | | Dr. Carragher Lab, The Scripps Research Institute |
| Software, algorithm | Relion 2 | *Scheres, 2012* | | Dr. Scheres Lab, MRC Lab of Molecular Biology |
| Software, algorithm | EMAN2 | *Tang et al., 2007* | | Dr. Ludtke Lab, Blue Mountain College |
| Software, algorithm | Chimera | *Pettersen et al., 2004* | | UCSF Resource for Biocomputing, Visualization, and Informatics |
| Software, algorithm | Modeller | *Webb and Sali, 2016* | | Dr. Sali Lab, University of CA San Francisco |
| Software, algorithm | Rosetta | *DiMaio et al., 2011* | | Dr. Baker Lab, University of WA |

## Bacterial strains

Transposon mutagenesis and subsequent chromosomal deletions of *mla* genes were performed in *Acinetobacter baumannii* ATCC 17978.

## A Mariner-based transposon vector for use in *Acinetobacter baumannii*:

To perform transposon mutagenesis a Mariner-based transposon vector was designed for use in *Acinetobacter baumannii* ATCC 17978. The new transposon vector, derived from pBT20, termed pMarKT, contains an outward facing pTac promotor as well as a selectable kanamycin resistance marker followed by an omega terminator within the Mariner arm sites (*Kulasekara et al., 2005*). The plasmid backbone contains the Mariner transposase gene C9 Himar, a *tetRA* resistance marker from Tn10, a p15A origin from pACYC184, and an oriT site for mobilization. The plasmid was constructed by PCR of select fragments followed by restriction digest and ligation of the cleaved ends. The new transposon vector was confirmed by restriction digest and partial sequencing.

## Transposon mutagenesis

Initial mutagenesis revealed that many hits occurred in the high affinity phosphate uptake transcriptional repressor *phoU* (A1S_0256). Subsequent rounds of mutagenesis were conducted on an ATCC 17978 *phoU* chromosomal deletion strain, and plated on high phosphate media to reduce the background level of cleavage of the chromogenic substrate. Chromosomal deletions were performed by allelic exchange using a pEX2tetRA vector, which was created by insertion of the *tetRA* tetracycline resistance marker from Tn10 into the pEXG2 plasmid (*Rietsch et al., 2005*). Roughly 1000 bp regions upstream and downstream of the genes of interest were amplified for homologous

recombination with the ATCC 17978 chromosome. Sucrose was used to counter-select against cells retaining the pEX2tetRA backbone, and deletions were confirmed by PCR. Complementation of deletions was accomplished by repairing the original deletion in the chromosome, again using the pEX system and allelic exchange.

Donor *E. coli* containing the pMarKT transposon vector were suspended in LB broth to an $OD_{600}$ of 40 and mixed with an equal volume of the recipient *A. baumannii* suspended to $OD_{600}$ of 20. 50 μL aliquots of this mixture were then plated in spots on a dried LB agar plate and incubated for 2 hr at 37°C (*Kulasekara et al., 2005*). Each 50 μL spot resulted in about 80,000 colonies of *A. baumannii* containing Mariner transposon insertions. The mutants were plated on LB agar containing 1X M63 salts, 50 μg/mL kanamycin, 30 μg/mL chloramphenicol, and 40 μg/mL XP substrate. Plates were incubated for at least 36 hr at 30°C to allow for the appearance of the blue color from cleavage of the XP substrate. Sequencing of the transposon insertions was adapted from the method described in *Chun et al. (1997)*, including semi-arbitrary two-step PCR amplification of transposon regions followed by sequencing.

## Ethidium bromide uptake assay

Bacteria were grown in 5 mL LB cultures to mid-log OD600 (0.3–0.6), then spun down and normalized in PBS to OD600 0.2. Prior to measurement, CCCP was added at 200 μM to inhibit the activity of efflux pumps. Ethidium bromide was added immediately prior to measurement to final concentration of 1.2 μM in 200 μL total reaction volume. Permeability was assessed using a PerkinElmer EnVision 2104 Multilabel Reader using a 531 nm excitation filter, 590 nm emission filter, and a 560 nm dichroic mirror. Readings were taken every 15 s for 30 min with samples assessed in triplicate in a Greiner bio-one 96-well flat bottom black plate.

## MIC measurements

MICs were determined in 96-well microtiter plates using a standard two-fold broth dilution method of antibiotics in LB broth. The wells were inoculated with $10^4$ bacteria per well, to a final well volume of 100 μL, and plates were incubated at 37°C with shaking unless stated otherwise. Experiments were performed thrice using two technical replicates per experiment. MICs were interpreted as the lowest antibiotic concentration for which the average $OD_{600}$ across replicates was less than 50% of the average $OD_{600}$ measurement without antibiotic.

## Crystal violet assay for exopolysaccharide production

Strains were inoculated to $OD_{600}$ 0.05 and grown overnight at 37°C in 2 mL LB broth with shaking in glass tubes. The next day, liquid was carefully decanted and the tubes left to dry for 2 hr at 37°C. Pellicles were stained with the addition of 0.1% crystal violet, then gently washed three times in $dH_2O$. Crystal violet was solubilized in a 80:20 solution of ethanol:acetone and read at 590 nm. P values were determined from a Student's t-test over three biological replicates per sample.

## MlaFEDB protein expression and purification

The *mlaFEDCB* operon from the genome of *A. baumannii* ATCC 17978 was subcloned into the pET-28a vector (Novagen, US) with a hexahistidine (−6HIS) tag fused at the C-terminus of the MlaB protein. The nucleotide sequence of the operon was confirmed using DNA sequencing. The plasmid was transformed into *E. coli* RosettaBlue strain. Cells were grown at 37°C in LB medium until the cell density reached an OD600 of 1.0. The temperature was then reduced to 16°C before induction with 1 mM isopropyl β-D-thiogalactoside (IPTG). After growth at 16°C for 18 hr, cells were harvested by using centrifugation at 4,200 g. Cells were resuspended in ice-cold buffer A (20 mM Tris-HCl (pH 8.0), 150 mM NaCl, 5% (v/v) glycerol) and subjected to three runs of homogenization at 10,000–15,000 psi using Avestin EmulsiFlex-C3 high pressure homogenizer (Avestin, Ottawa, Ontario, Canada). The homogenate was centrifuged at 17,000 g for 10 min at 4°C, and then the supernatant was ultra-centrifuged at 100,000 g for 60 min. The membrane fraction was resuspended in buffer A supplemented with 1% (w/v) dodecyl-β-d-maltopyranoside (DDM) and was slowly stirred for 1 hr at 4°C. After another ultra-centrifugation at 100,000 g for 30 min, the supernatant was collected and loaded on 2 ml of $Ni^{2+}$-nitrilotriacetate affinity resin (Ni-NTA from Qiagen, Germany) pre-equilibrated with buffer A supplemented with 5 mM imidazole and 0.025% (w/v) DDM. After incubating for 1 hr, the

resin was washed with 50 ml buffer A supplemented with 20 mM imidazole and 0.025% (w/v) DDM. The protein sample was eluted with 10 ml elution buffer containing buffer A, 300 mM imidazole, and 0.025% (w/v) DDM, and was concentrated to 0.5 ml. The concentrated protein sample was then loaded onto a Superdex-200 column (10/30, GE Healthcare, US) pre-equilibrated with 20 mM Hepes (pH 7.0) 150 mM Nacl 0.025% DDM. Peak fractions were collected and the pooled protein sample was concentrated to 1 mg/ml.

## Cryo-EM sample preparation, data acquisition and image processing

Purified Mla complex at ~1 mg/ml was applied to glow-discharged holey grids, blotted for 6 s, and plunged in liquid ethane using a Vitrobot (FEI). Images were acquired on a FEI Tecnai G2 F20 200 kV Cryo-TEM equipped with a Gatan K-2 Summit Direct Electron Detector camera with a pixel size of 1.26 Å/pixel. 500 micrographs were collected using Leginon (*Suloway et al., 2005*) spanning a defocus range of −1 to −2 µm.

Movie frames were aligned with MotionCorr2 (*Zheng et al., 2017*) and the defocus parameters were estimated with CTFFIND4 (49). 333 high-quality micrographs were selected by manual inspection, from which ~ 55,000 particles were picked with DOG in Appion (*Lander et al., 2009*). Particle stacks were generated in Appion using a box size of 200 pixels. Several successive rounds of 2D and 3D classification were performed in Relion 2 (*Scheres, 2012*; *Kimanius et al., 2016*) using an initial model generated by Common Lines in EMAN2 (53) leading to a final stack of ~14,000 particles for 3D structure refinement in Relion.

## Structure modeling and docking in the EM density

The structures of MlaB and MlaF were modeled using the threading server Phyre (*Kelley et al., 2015*) based on the structures of the anti-sigma factor antagonist tm1081 (PDB ID 3F43, 18% sequence identity to MlaB) and the ABC ATPase ABC2 (PDB ID 1OXT, 36% sequence identity to MlaF) respectively. Two copies of each structural model were positioned in their putative location within the EM map using Chimera (*Pettersen et al., 2004*) and their position was optimised using the Fit to EM map option. The abMlaD and abMlaE structures were modelled on ecMlaD and ecMlaE structural models deposited in the Gremelin database (*Ovchinnikov et al., 2017*), using Modeller. For abMlaD, the N-terminal TM helix and the insert region were modelled ab initio using the Rosetta suite (*DiMaio et al., 2011*) and positioned in their putative localisation in Chimera. The MlaD hexamer, as well as the MlaE dimer, were modelled with Rosetta using a EM-guided symmetry modelling approach described previously (*Bergeron et al., 2013*). The final model was refined with Rosetta.

## Membrane isolation and separation

Cells were resuspended in 20 mL of 0.5 M sucrose, 10 mM Tris pH 7.8, 75 µg freshly prepared lysozyme (Roche 10837059001), and 20 mL of 0.5 mM EDTA, and kept on ice with gentle stirring for 20 min. Samples were homogenized (Avestin EmulsiFlex-C3) and spun down at 17,000 g for 10 min to removed un-lysed cells prior to ultracentrifugation. Membranes were spun down using a Ti45 Beckman rotor at 100,000 g for 1 hr and then added to the top of a sucrose gradient. IM and OM were separated by 18 hr ultracentrifugation using a SW-41 rotor in a Beckman Coulter Optima L90X ultracentrifuge. Spheroplast formation and sucrose gradient separation of IM and OM was adapted from a method by *Osborn et al. (1972)* by use of a defined 73%–53–20% sucrose gradient as described in *Dalebroux et al. (2015)*. Our sucrose gradients contain three distinct concentrations of sucrose, and inner and outer membranes separate into distinct bands that are collected individually (*Figure 5—figure supplement 3F*). To limit any potential mixing of the membranes, the inner membrane is collected from the top of the tube while the outer membrane is collected by puncturing the bottom of the tube and allowing the bottom band to be collected. The purity of membrane separation by this method was confirmed by NADH assay and by Western blotting for the *A. baumannii* OM-localized OmpA protein, with 10 µg of total protein loaded into each lane as measured by Bradford protein assay (*Figure 5—figure supplement 3*).

## GPL extraction and TLC

GPLs from isolated membranes were extracted using a 0.8:1:2 ratio of water: chloroform: methanol as per the method of Bligh and Dyer (*Bligh and Dyer, 1959*). Two-dimensional TLC was performed using silica gel 60 plates and immersion in Solvent System A (60:25:4 $CHCl_3$:$CH_3OH$:$H_2O$), followed by Solvent System B (80:12:15:4 $CHCl_3$:$CH_3OH$:$CH_3COOH$:$H_2O$) in the orthogonal direction.

## MlaC and MlaD protein purification and GPL extraction

Primers were designed to amplify the *mlaC* gene of ATCC 17978, excluding the signal sequence for export from the cytoplasm, and the periplasmic domain of *mlaD* of ATCC 17978, excluding the membrane-spanning domain. These fragments were cloned into pET29b and expressed with a car-boxy-terminal hexahistidine (−6HIS) tag in BL21 *E. coli* with 2 hr induction. Cells were pelleted and resuspended in Tris-buffered saline containing 10% glycerol (TBSG) and protease inhibitor cocktail (Roche, Complete EDTA-free). Cells were lysed by homogenization (Avestin) and ultracentrifuged at 100,000 g for 1 hr to spin down membranes. The supernatants were then applied to a 5 mL-HiTrap (TM) Chelating HP Ni-affinity column pre-loaded with 0.1 M $NiSO_4$ and equilibrated with TBSG. The proteins were eluted from the column using FPLC (Akta) by applying a stepwise gradient of 25 mM, 50 mM, and finally 300 mM imidazole for protein elution. Elution was monitored by UV-absorption at 280 nm. The MlaC- and MlaD-containing fractions were then further purified by injecting into a HiLoad 120 ml-6/600 Superdex(TM) 200 preparative grade size-exclusion column equilibrated in TBSG using a flow rate of 1 mL/min. The purity of the collected protein fractions was confirmed by SDS polyacrylamide gel electrophoresis. Proteins were diluted to 2 mg/mL and dialyzed overnight in 1 L TBSG at 4°C with stirring. GPLs were extracted from 1 mg each of purified proteins MlaC and MlaD by the method of Bligh and Dyer and analyzed by LC-MS/MS as previously described.

## Lc-ms/MS

Retention of PG, CL, PE, and Lyso-CL was achieved at a flow rate of 0.3 mL/min using mobile phase A [$CHCl_3$/$CH_3OH$/$NH_4OH$ (800:195:5 v/v/v)] and mobile phase B [$CHCl_3$/$CH_3OH$/$NH_4OH$ (600:340:5 v/v/v)]. The chromatography method used is a three-step gradient as described in the SI Materials and methods of *Dalebroux et al. (2014)*. The samples were run on an Agilent Zorbax Rx-SIL silica column (2.1 × 100 mm, 1.8-Micron) using an Agilent HPLC autosampler. Mass spectrometry was performed using an AB Sciex API4000 Qtrap with multiple reaction monitoring (MRM). The identities of the major GPLs present in the *A. baumannii* membrane were predicted by parent ion scans.

## Stable isotope assay development

The Q1/Q3 transitions of glycerolphospholipids from cells grown in 2-$^{13}C$ acetate were determined using a Thermo Orbitrap LTQ. The integrated peak areas of both $^{13}C$-labeled and unlabeled GPLs from the AB Sciex API4000 Qtrap were used to calculate the ion-current ratios for each GPL species. The ratio of labeled GPL for each unique species can be calculated based on the following equation:

$R_{lab} = R_i R_b$ (*MacCoss et al., 2001*)

Where $R_i$ is the ion-current ratio of labeled GPL to unlabeled GPL within the sample and $R_b$ is the ion-current ratio of samples before the administration of the tracer, $^{13}C$-acetate, and represents the natural background abundance of the stable isotope species within the bacterial membrane. $R_{lab}$ approximates the molar ratio of labeled species to unlabeled species ($n_{lab}/n_{un}$) according to the equation ($n_{lab}/n_{un}$) = [$R_i$-$R_b$]/k, where k is the molar response factor of the instrument and is ideally equal to unity (*MacCoss et al., 2001*).

To demonstrate that OM phospholipases will not distinguish between labeled and unlabeled GPL and therefore will not affect the ratio of labeled to unlabeled GPL obtained from this assay, we compared ratios of labeled and unlabeled GPL from wild type *A. baumannii* and deletion mutants in *pldA*. Bacteria were grown carrying either the empty pMMB::*kan* vector, or expressing the Walker box mutant MlaF$^{K55L}$. Accumulation of newly synthesized GPL was observed in those strains expressing MlaF$^{K55L}$ when compared to the vector control, across various species of GPL. Of strains expressing the vector control, on average 51.84 ± 1.07% and 52.07 ± 1.23% of newly synthesized PG C16:0/18:1 appeared on the inner membrane of wild type and Δ*pldA*, respectively, after one hour incubation with $^{13}C$ acetate, while 66.33 ± 1.23% and 62.60 ± 1.70% of newly synthesized PG C16:0/18:1 accumulated at the inner membranes of wild type and Δ*pldA* expressing MlaF$^{K55L}$. In vector controls

strains, 48.53 ± 1.37% and 51.01 ± 0.55% of newly synthesized PG C16:0/16:0 appeared on the inner membrane of wild type and $\Delta pldA$, respectively, after one hour incubation with $^{13}$C acetate, while 62.98 ± 1.01% and 60.41 ± 1.25% of newly synthesized PG C16:0/16:0 accumulated at the inner membranes of wild type and $\Delta pldA$ expressing MlaF$^{K55L}$. In vector controls strains, 50.17 ± 1.31% and 50.49 ± 1.15% of newly synthesized PE C16:0/18:1 appeared on the inner membrane of wild type and $\Delta pldA$, respectively, while 60.14 ± 0.93% and 62.06 ± 1.07% of newly synthesized PE C16:0/18:1 accumulated at the inner membranes of wild type and $\Delta pldA$ expressing MlaF$^{K55L}$.

### Stable isotope GPL analysis and culture conditions

Cultures of *A. baumannii* ATCC 17978 were grown in M63 media containing 5 mM sodium acetate and 4 mM MgCl$_2$ to OD600 0.4, then washed and resuspended in media containing 5 mM 2-$^{13}$C sodium acetate (Cat. No. CLM-381–0, Cambridge Isotope Laboratories, Inc.). Membrane fractions were isolated from both wild type and *mla* mutant *A. baumannii* at simultaneous time points, and GPL were extracted and assessed using previously established LC-MS/MS methods with additional MRM values to account for the increased m/z ratios of $^{13}$C-labeled GPL. MRMs were selected to account for PG and PE having acyl chains of either C16:0/16:0, C16:0/18:1, and C18:1/18:1 as these were determined by total ion scan MS to be the predominant species of PG and PE GPL. Pulse experiments were performed at least twice for each mutant.

## Acknowledgements

We thank Dale Whittington and Dr. Scott Edgar at the Mass Spectrometry Center, Department of Medicinal Chemistry, University of Washington for technical help with MS analysis; and Mauna Edrozo for technical help. This work was supported by a grant from NIAID, U19AI107775.

## Additional information

### Funding

| Funder | Grant reference number | Author |
| --- | --- | --- |
| National Institute of Allergy and Infectious Diseases | U19AI107775 | Samuel I Miller |
| National Institutes of Health | R01GM118396 | Justin M Kollman |

The funders had no role in study design, data collection and interpretation, or the decision to submit the work for publication.

### Author contributions

Cassandra Kamischke, Conceptualization, Data curation, Formal analysis, Investigation, Visualization, Methodology, Writing—original draft, Writing—review and editing; Junping Fan, Data curation, Investigation, Visualization, Writing—review and editing; Julien Bergeron, Data curation, Formal analysis, Investigation, Visualization, Writing—original draft, Writing—review and editing; Hemantha D Kulasekara, Formal analysis, Supervision, Investigation, Methodology, Writing—review and editing; Zachary D Dalebroux, Data curation, Supervision, Investigation, Methodology; Anika Burrell, Data curation, Formal analysis, Investigation; Justin M Kollman, Supervision, Funding acquisition; Samuel I Miller, Conceptualization, Supervision, Funding acquisition, Project administration, Writing—review and editing

### Author ORCIDs

Samuel I Miller [iD] https://orcid.org/0000-0003-1638-2181

### Decision letter and Author response

Decision letter https://doi.org/10.7554/eLife.40171.029
Author response https://doi.org/10.7554/eLife.40171.030

# Additional files

## Supplementary files
• Supplementary file 1. Results of transposon mutagenesis screen for genes involved in outer membrane barrier function in ATCC 17978.
DOI: https://doi.org/10.7554/eLife.40171.021

• Supplementary file 2. Accumulation of newly synthesized glycerophospholipids in *A. baumannii* inner and outer membranes.
DOI: https://doi.org/10.7554/eLife.40171.022

• Transparent reporting form
DOI: https://doi.org/10.7554/eLife.40171.023

## Data availability
The cryo-EM map has been deposited in the Electron Microscopy Data Bank with accession code EMD-8738 (8.7 Å map). The coordinates for the MlaBDEF model have been deposited to PDB, accession code 6IC4.

The following datasets were generated:

| Author(s) | Year | Dataset title | Dataset URL | Database and Identifier |
|---|---|---|---|---|
| Kamischke C, Fan J, Bergeron J | 2018 | 8.7 Å cryo-EM map | https://www.ebi.ac.uk/pdbe/entry/emdb/EMD-8738 | Electron Microscopy Data Bank, EMD-8738 |
| Kamischke C, Fan J | 2018 | Coordinates for the MlaBDEF model | http://www.rcsb.org/structure/6IC4 | Protein Data Bank, 6IC4 |

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
