## [Decision Letter]

[Editors’ note: a previous version of this study was rejected after peer review, but the authors submitted for reconsideration. The first decision letter after peer review is shown below.]

Thank you for submitting your work entitled "An ABC transporter delivers glycerophospholipids to the *Acinetobacter baumannii* outer membrane" for consideration by *eLife*. Your article has been reviewed by three peer reviewers, and the evaluation has been overseen by a Reviewing Editor and a Senior Editor. The reviewers have opted to remain anonymous.

Our decision has been reached after consultation between the reviewers. Based on these discussions and the individual reviews below, we regret to inform you that your work will not be considered further for publication in *eLife*.

While all reviewers agree that the manuscript addresses an important topic, they all think that the evidence provided is not sufficient to support the main claim of the manuscript, that the Gpt system delivers phospholipids to the outer membrane (OM). This is an especially sensitive issue because the *E. coli* ortholog Mla complex has been described to function in the opposite direction from the OM to the IM. While the reviewers do not exclude that your conclusions are correct, they all think that this controversy can only be resolved if a large body of additional experiments is provided. Briefly, clear membrane fractionation controls should be provided, but it should be explained how the PL decrease in the OM is compensated. Other approaches such as EM and live microscopy would also strengthen the conclusions.

The policy at *eLife* is to only allow revisions of manuscript that could be performed within a two-months time period. Given, the amount of perceived additional experiments we have are rejecting the manuscript. The individual reviews are appended below to help prepare the manuscript for submission elsewhere.

*Reviewer #1:*

This work by Kamischke et al. presents the interesting finding of a glycerolipid transport system, which appears to have a major role in establishing the composition of the OM.

The transporter is present in other gram negative bacteria and the finding would thus be an important insight into the mechanism of OM biogenesis.

The work nicely provides genetic, structural and biochemical assays to support the authors conclusions

Overall, I am enthusiastic about the work, but I have a number of concerns.

– First, the discovery of the Gpt system is not exactly novel, the Mla complex, apparently the direct homolog, has been previously identified in *E. coli* (which the author acknowledge). However contrary to previous interpretations, Gpt is proposed to transport glycerolipids from the IM to the OM while Mla (f I understand correctly) was proposed to maintain lipid asymmetry in the OM, transporting lipids from the OM to the IM. I am not an expert in this field but because this creates a controversy, I am wondering whether the data presented is sufficient to make this claim. The conclusion mostly comes from the assay presented in Figure 7 showing that newly synthesized phospholipids accumulate at the IM in *gpt* mutants. However, I do not understand why the OM composition appears mostly unaffected in the same mutants in that assay? Also given that the authors have this assay in hand it should not be too complicated to test the *E. coli mlA* mutant and see if they get similar results? That would help solving this controversy.

Bottom line, this part should be absolutely convincing for the paper to be accepted.

– Some microscopy of the *gpt* mutants (EM, phase contrast) would be helpful to describe the phenotypes in these mutants. Could permeability of the mutants be shown directly in individual cells?

– Statistics are missing in several instances especially in in Figure 1 and Figure 7 which only shows representative experiments.

*Reviewer #2:*

This work deals with the most poorly understood aspect of envelope biogenesis in Gram-negative bacteria: transport of phospholipids (PLs) between the inner membrane (IM) and outer membrane (OM). Kamischke et al. characterize the Gpt system of Acinetobacter baumannii, which is orthologous to the previously identified Mla system in *E. coli*. A portion of this work describes how some of what we know about Mla also applies to Gpt: *mla/gpt* mutants have OM permeability defects; IM components of Mla/Gpt system form a complex; part of the Mla/Gpt system is a predicted ABC transporter and activity of its ATPase is required for transport; and finally, some periplasmic components of Mla/Gpt bind PLs.

The remaining part of this manuscript describes two novel findings but raises the following serious concerns:

1) Cryo-EM structural information of the GptBDEF IM complex. This is a novel structure that is only validated by the fitting of Phyre model structures of the cytoplasmic components (GptBF) and previous data showing that MlaD (GptD) is hexameric. Because the structure is of low-resolution (~11 Å), its impact is minimal – it does not provide insight into the mode of function or guidance for future work on function.

2) The authors conclude that Gpt transports PLs from the IM to OM. This is very surprising because both the Mla system of *E. coli* and its chloroplast orthologous system have been reported to function in retrograde PL transport based on in vivo data. Previously, Malinverni and Silhavy (reference #8) concluded that Mla functions in retrograde transport because they obtained in vivo data demonstrating that PLs accumulate in the outer leaflet of the OM: in *mla* mutants, PagP was activated and defects were suppressed by PldA overexpression. It is impossible to explain those data in a model where Mla transports PLs from the IM to the OM, as it is suggested here for Gpt. It is also very difficult to imagine how a multi-protein transport system that is driven by an ABC transporter to transport PLs from one membrane to another could have evolved to run in opposite directions in different organisms.

Here, Kamischke et al. conclude that Gpt is involved in anterograde transport because of data presented in Figures 6 and 7, where the authors measure PL content in the IM and OM. These data rely on the separation of IM and OM using a density ultracentrifugation procedure. How well these two membranes were separated is crucial to validate this work. Therefore, the authors should show the data they mentioned in the Materials and methods that illustrate proper separation of the IM and OM. In fact, the authors should at the very least show stained protein gels (not just immunoblots) of all fractions so that readers can assess the fractionation procedure. Furthermore, I worry that this technique might have misled the authors because of the following issue. Separation of IM and OM only works when the density values of these membranes are different enough. There are two factors contributing to this difference: the different protein:lipid ratio between the two membranes and the LOS:phospholipid ratio in the OM. If the Gpt system was responsible for retrograde transport, as previously proposed for Mla, a *gpt* mutant would accumulate PLs in the OM. This would cause the density of the OM (or domains/areas in the OM) to be more IM-like, making the fractionation more difficult or even impossible depending on the extent of the defect. In fact, it is likely that if PLs accumulated in the OM, the OM would be a mixture of two types of bilayers: some areas would be like wild-type OM (asymmetric bilayer containing both LOS and PLs), while other areas would be IM-like (symmetric PL bilayer). That is, some OM domains (or areas) would fractionate as expected (heavy density), while others would fractionate with the IM or somewhere in between the OM and IM fraction (lighter density). If this happened, fractionations would appear normal except that one would expect excess PLs in the IM, compared to the OM. This is what the authors' data show, so I wonder if they are misinterpreting the data. Can the authors rule out this possibility? Furthermore, the authors use a fractionation procedure that is based on spheroplasting cells, which leaves domains/areas of the OM still connected to the IM via trans-envelope complexes (e. g. Tefsen J Biol Chem. 2005). Would the same results be obtained if cells were lysed by high pressure?

Another issue that their data raises is this: The OM should cover more surface area than the IM. Here, the authors say that the OM of a *gpt* mutant has a lot less PLs than the OM of the wild type and that the IM of the mutant has more PLs than that of the wild type. Their data suggest that the OM in the *gpt* mutant covers less surface area that in IM. How is this possible? What is filling the void in the OM caused by the decrease in PLs? The authors do not provide any information about the overall composition of the OM in *gpt* mutants that could explain how it can have less PLs while the IM has more. Again, different composition of the OM could affect how it (or domains of it) fractionates.

The authors should also address these additional major problems:

3) Why isn't Figure 6B showing accumulation of PG and PE in the IM? These data seem to contradict the model. Even Figure 6A does not appear to show much of an increase of PLs at the IM.

4) The authors state (without providing data) that cardiolipin is made at the OM. If that is true, based on the authors' model (Gpt transports the substrates needed to make cardiolipin at the OM), one would expect that the overall cardiolipin content in *gpt* mutants would be very low, especially in comparison to the other PL species. That is not what Figure 6A shows. Why?

5) Could the authors please show error bars for Figure 7?

6) Discussion section: "While it is possible that the Gpt system in *E. coli* serves a different primary function than in *A. baumannii*, the phenotypes observed in *E. coli mla* mutants might also be explained by a disruption of outer membrane structure stemming from decreased concentrations of outer membrane PLs, leading to activation of the PagP enzyme." This weak argument does not make any sense. PagP is activated by an increase in PLs at the outer leaflet of the OM.

*Reviewer #3:*

The article is written and presented in a seamless manner, where figures, hypotheses, and figures are clearly articulated. This is a testament to the overall quality of the work and presentation. However, this is not to say that it has no weaknesses, some of which need to be addressed prior to publication. The authors' conclusions are never heavy-handed or written in absolute terms; as such they accommodate alternative opinions and possibilities; especially important, as their final figure, serves to upturn the existing dogma in the field, regarding the function of a homologous operon best studied in *E. coli*. Some might argue that renaming these homologs is unnecessary. The authors should strongly consider their proposal to rename these genes for the following reasons: i) the buried mention (Materials and methods) that cardiolipin is not transported by this system, argues that it is not in fact a general "glycerophospholipid" transporter (see below additional comments on cardiolipin); ii) these authors have only demonstrated in one strain of *A. baumannii that* some degree of PE and PG transport to the OM likely occurs as a consequence of this operon, and have not overturned the likely function of this operon to maintain asymmetry of the outer membrane. It's possible that context, such as organism or environments, favors utilization of this machinery in one direction or another. Also of great importance is the issue as to what fills in for the loss of GPL in the outer membrane as it is highly unlikely that it is LOS at the inner leaflet of the outer membrane. The authors' consideration on these matters, and their ability to adequately address these concerns will likely impact publication.

Major concerns/comments:

1) As noted above the Mla (Gpt) system has been previously characterized in *E. coli* and hypothesized as a retrograde GPL transporter. Clearly the *mla* system is not required for growth, so if it is functioning as an anterograde transporter what is replacing GPLs in the inner leaflet of the OM in *mla* mutants. In the view of this reviewer, the burden of proof is now higher given the previous work on the *E. coli mla* system.

2) The audience must be provided with inner membrane and outer membrane separation data throughout the manuscript. This is critical as the reader must be convinced that changing the ratio of GPL/LOS/OM protein allows for proper separation of *Acinetobacter* membranes.

3) Do the authors have data to demonstrate that the K55L "dominant negative" mutation truly abrogates ATP hydrolysis, since it is a critical assumption in interpreting the data in Figures 1 and 2?

4) Could the authors please speculate why only A1S_3102 and A1S_3103 were identified using BCIP-Toluidine screen of Tn library? Where there not Tn insertions in the other members of the operon?

5) Why do the authors not discuss MlaA and why was a mutant not made in this gene? Also, it is not labeled in the model? Does it not function in the pathway in *Acinetobacter*?

6) Figure 6. Several comments on this figure:

a) In 6A – it appears there are more GPLs by TLC in the OM of the wild type which is odd

b) In 6B, – if *gpt* was an anterograde transporter one would expect to see accumulation of GPLs in the inner membrane and decreases of GPLs in the outer membrane. When you look at the bar graph, the levels of PE and PG don't increase in the inner membrane in the *gptC* mutant and they actually decrease a little. Why?

7) Have the authors performed electron microscopy to see if they observe changes in the inner membrane (membrane invaginations) because of loss of *gpt* mediated GPL transport? This sort of data would greatly strengthen their hypothesis.

8) Discussion paragraph two: The Mla system was previously reported as a retrograde transport system for *E. coli* and in *mla* mutants PagP is activated. Since PagP is only functional when PLs are present at the surface, the idea was that in *mla* mutants GPLs are mislocalized in the outer leaflet of the OM. In the discussion, the authors rationalize that the phenotypes observed in *E. coli mla* mutants might also occur because GPL transport to the OM is not as efficient and because of this the LPS-to-GPL rations are off, resulting in flipping of GPLs to the outer leaflet and this is why PagP is activated. Is this likely, as PagP needs GPLs for activity and in the authors' model the Gpt (Mla) system is responsible, at least in part, for transport of GPLs to the OM?

9) Discussion paragraph three: The authors note that since the Mla (Gpt) system is upregulated in LOS deficient Acinetobacter, it suggests that Mla serves as a anterograde transporter as increased transport of GPLs may serve to compensate for lack of LOS in the outer membrane. Why couldn't it be the reverse, that the Mla system is upregulated in order to prevent too many GPLs in the OM. This goes back to what fills in the gaps when LOS is missing. After searching on PubMed, this reviewer found a recent paper from Boll et al., reporting that key lipoproteins seemed to be present in the outer leaflet of LOS deficient *Acinetobacter*.

[Editors’ note: what now follows is the decision letter after the authors submitted for further consideration. They were asked to provide a plan for revisions before the editors issued a final decision.]

Thank you for sending your article entitled "The *Acinetobacter baumannii* Mla system and glycerophospholipid transport to the outer membrane" for peer review at *eLife*. Your article is being evaluated by two peer reviewers, and the evaluation is being overseen by a Reviewing Editor and Wendy Garrett as the Senior Editor.

Given the list of essential revisions, including new experiments, the editors and reviewers invite you to respond within the next two weeks with an action plan and timetable for the completion of the additional work. We plan to share your responses with the reviewers and then issue a binding recommendation.

The current manuscript proposes that the *Acinetobacter baumannii* Mla system transports glycerophospholipid to the outer membrane. This statement challenges a large body of literature on both the *E. coli* and *A. Baumannii* systems (and to some extent *V. Cholerae* and *H. influenza*), all of which favor that Mla is instead involved in OM turnover and retrograde transport. After consultation with the reviewers, several key points would have to be clarified for publication in *eLife*:

1) The reference to existing literature must be improved in particular to reconcile and potentially explain the differences between this study and published studies. If needed, additional experiments must be provided to resolve/address these conflicts. Along these lines:

– Powers and Trent, 2018, published evidence for retrograde transport in the same organism, which is not commented by the authors.

– Contrarily to this study, multiple laboratories have reported that deletion of *mlA* genes is not associated with increased sensitivity to vancomycin and in fact linked to vancomycin resistance gene induction.

– Roier et al. (Nat Comm, 2016) have shown that *mlA* deletion is linked to OMV production in *V. Cholerae* and *H. influenza*, which is not taken into account in this study. Can the authors rule out that lower GPL levels in the OM are not due to the loss of OM materials?

2) The pulse chase assay is an important innovation of this study and the major experimental evidence to support the proposed directionality of Mla. Proving the consistency of this assay is therefore paramount to support the authors’ conclusions. The authors need to provide all the controls confirming that the IM and the OM were correctly separated in sucrose density gradients, which apparently is performed for the first time in *A. Baumannii*. For this, sucrose gradients need to be shown, with a complete set of IM and OM controls. A series of helpful recommendations is provided in the individual reviews. Variations in the assay starting point and in particular the high variance in total labeled lipids should also be explained.

*Reviewer #2:*

In "The *Acinetobacter baumannii* Mla system and glycerophospholipid transport to the outer membrane," Kamischke et al. report that the previously characterized Mla system functions in an anterograde manner. This is a well-written manuscript and the subject matter is important. The authors used a transposon library to identify genes required for OM integrity in *A. baumannii* by monitoring for release of an endogenous periplasmic phosphatase. Among the 58 genes identified in the screen, two genes encoded for components of the Mla system. The finding that mutations in *mla* genes results in OM permeability defects in Ab aligns with previous data reported in *E. coli* and other Gram-negative organisms (e.g. *Vibrio cholerae, Haemophilus influenza*). Previously, the Silhavy Laboratory demonstrated that Mla mutants show increased sensitivity to SDS-EDTA and deletion of any of the Mla components resulted in mislocalization of phospholipids to the outer leaflet of the outer membrane. It was found that the outer membrane phospholipase, PldA, was a suppressor of the *mla* mutant SDS-EDTA sensitivity (see below for additional details on previous work). This is a key point as it suggests that the role of the Mla system is to remove phospholipids from the outer leaflet.

In the current work, the authors went on to characterize the Ab Mla inner membrane complex via cyro-EM. In the view of this reviewer, this is the most substantial contribution of the current work. Previously, Ekiert and co-workers characterized multiple MCE-domain containing complexes, including the Mla complex (Cell, 2017), and the work completed here is a nice addition to the field. Unfortunately, the other findings reported here are highly problematic. The authors reach the conclusion that Mla functions as an anterograde phospholipid transporter, which is in direct conflict with previous work from multiple groups suggesting that Mla functions in a retrograde manner including a recent paper in *A. baumannii* (Powers et al., 2018). The Mla mutants presented here have very different phenotypes than those reported for other Gram-negatives and the biochemical data to support the anterograde claim are not convincing for various reasons.

Major Points:

1) Kamischke and co-workers report that deletion of *mlaC* or *mlaF* results in increased sensitivity to vancomycin (~10 fold). First, this goes against previous literature, reported by multiple labs, that *mla* mutants do not show increased sensitivity to vancomycin (or other antibiotics such as novobiocin). Actually, according to others the level of vanc resistance actually increases upon deletion of *mla* genes. How do the authors explain these differences?

2) Subsection “The Mla system is necessary for *A. baumannii* OM integrity”: What is the rationale for authors to complement mutations in original locus? This does not address the possibility of off-target gene expression alterations. This is even more relevant since it has already been reported that previous *mla* insertational mutants show polar effects (Silhavy, 2009).

3) Figure 4 and 5 and supplemental: Authors do not adequately illustrate the effectiveness of their membrane separation. It is established that many diderm organisms cannot be physically fractionated into OM and IM. Furthermore, to the reviewer's knowledge, this is the first publication to include membrane separations for *Acinetobacter baumannii*. The authors need to include significantly more information to assure the readers that their membrane fractionation works robustly in *A. baumannii*. Currently, the authors only include an outer membrane marker for an unknown fraction of the gradients. This is insufficient in the context of this work. The authors should include an inner membrane marker and an outer membrane marker across the gradient. Also, protein distribution and turbidity measures over the entire gradient would be very helpful.

4) Figure 4B: Where do the authors hypothesize the GPLs go? They presumably see a decrease in PG and PE in the outer membrane, but no subsequent increase in the inner membrane at steady state. Could this not be explained by vesicular blebbing, which would be an expected consequence after deletion of a retrograde transport system? In fact, an argument could be made that these findings are entirely an artifact of vesicular blebbing in the absence of Mla. In fact, Roier et al. have shown that deletion of *mla* components in both *V. cholerae* and *H. influenza* results in increased outer membrane vesicles. However, the authors do not consider this option or discuss this previous work.

5) Also, in reference to the Roier paper. In the current work the authors make the claim that "biochemical analysis for the membrane GPL composition for Mla mutants has not been published for any organism" This is not correct and these data were reported by Roier et al. Also, Roier did not observe the drastic reduction in OM phospholipid abundance reported in the current manuscript.

6) Discussion paragraph three: Authors suggest the function of MlaA is to allow MlaC to deposit phospholipids into the periplasmic-leaflet of the OM. The structure of MlaA identified a loop, preventing periplasmic-leaflet phospholipids from entering the MlaA channel. MlaA* lacks this loop, resulting in periplasmic-leaflet phospholipids entry into the MlaA channel. The result is a gain of function increased sensitivity to membrane stress.

7) Figure 5 and Figure 5—figure supplement 2: The authors devised a clever pulse chase assay. Why do the authors see such different starting points for the labeled cultures? Based on the variability of labeling by lipid species (even for the wildtype cultures), there is a concern regarding the consistency of the method.

8) Supplementary file 2: Furthering the concern about the pulse chase assay is the data presented in Supplementary file 2. We analyzed 3 replicates of the data provided, and it appears that the total lipid labeled achieved by this pulse chase assay is highly variable. How do the authors feel confident in drawing the conclusions presented when consistent data cannot be achieved with wild-type? In some replicates, entire lipid species were unable to be detected. How are the figures presented using this data representative, when the raw data is not?

*Reviewer #3:*

The outer membrane (OM) of Gram-negative bacteria is an asymmetric lipid bilayer, composed of an inner leaflet of glycerophospholipids (GPL) and an outer leaflet of lipopolysaccharides (LPS). Whereas we now have a reasonable understanding of how LPS molecules are transported across the periplasm and inserted into the OM, we still do not know how GPL molecules are transported from the inner to the outer membrane. In this article, Kamischkke et al. propose an answer to this long-standing question.

First, the authors carried out a genetic screen to identify genes important for the integrity of the OM in *Acinetobacter baumannii*, an opportunistic pathogen. They screened a library of transposon mutants for colonies with lesions in the OM, reasoning that bacteria with an impaired OM will allow the chromogenic substrate XP to reach the periplasm where it can be degraded by a phosphatase, yielding a blue color. The results of the screen were validated using different assays. This led them to identify mutants with transposon insertions in genes encoding core components of the Mla system. The Mla system has been identified in *E. coli* where it was proposed to play a role in OM maintenance, transporting GPL from the OM back to the IM. The authors also confirmed that the phenotype exhibited by the mutants resulted from a lack of transport function and then used cryoEM to characterize the structure of the MlaBDEF complex. They also confirmed that components of the system interact with GPL.

The work described in this first part of the manuscript is convincing, the results are solid and very well presented. However, this first part mostly confirms work previously carried out in *E. coli* where similar screens were performed and where the Mla system was characterized, including at the structural level. In particular, the structure of the *E. coli* Mla complex was recently reported, although at a lower resolution. Overall, this part of the work is interesting and carefully done, but its novelty is not very high.

The second part of the article contains what I think could warrant publication in *eLife*. Indeed, the authors provide experimental evidence that the Mla system transports GPL from the IM to the OM, which would make of this system the first identified transporting GPL in this direction. This conclusion is based on a number of experiments. First, analysing IM and OM fractions separated by density centrifugation with TLC and LC-MS/MS, they showed that GPL levels were reduced in the OM of the *∆mlaC* mutant. Second, they designed an MS-based strategy to quantitatively follow intermembrane transport of radiolabelled GPL in membrane fractions and showed that GPL accumulate in the IM of Mla mutants.

These results are extremely interesting and are consistent with the model that the Mla system transports GPL from the IM to the OM, which would represent a major finding. However, I have a number of important concerns that need to be addressed.

1) Given that the authors quantify GPL in membrane fractions, they need to provide all the controls confirming that the IM and the OM were correctly separated in sucrose density gradients. Only one control, showing the localization of OmpA in the OM fraction is provided. I think that more fractions of the sucrose gradients need to be shown, with additional IM and OM controls. My concern is that the accumulation of lipids in one membrane could have an impact on how the two membranes are separated and therefore on the results of the quantitative analysis. In addition, if Mla mutants have portions of the OM with GPL both in the inner and outer leaflet, where will these "GPL bilayers" be found in the sucrose gradients?

2) If GPL accumulate in the IM, what is the impact of this accumulation on the IM? A thorough microscopy analysis should also be carried out, which would strengthen the conclusions. Does the total surface area of the IM increase as a result of lipid accumulation?

3) The authors should also verify that the lower GPL levels in the OM do not result from the loss of OM material via increased vesiculation.

Providing these data is even more important that the direction proposed here for GPL transport is in contradiction with the work performed by others indicating that the Mla system is involved in the retrograde transport of GPL from the OM to the IM. In this regard, the authors need to take into account the recent publication by Powers and Trent in PNAS (2018) providing "strong biological evidence" for the retrograde transport of GPL by the Mla system in the same bacterium (*A. baumannii*). They need to explain how these data are compatible with theirs.

[Editors’ note: formal revisions were requested, following approval of the authors’ plan of action.]

Thank you for your careful response to the reviewers' comments. After further consultation with the reviewers we decided that your response could adequately address the comments provided that in addition to the proposed changes the revision further contains the following new experiments:

1) Given the existing literature, we would like further that the discrepancy on the vancomycin sensitivity be resolved by verification of this phenotype in a different *A. baumannii* strain.

2) Completion of sucrose gradients with the proper controls to verify inner and outer membrane separation for both wild type and *mla* mutants (e.g. NADH oxidase assays). A clear description of the gradient methodology and images showing separation of the membrane as provided in the rebuttal letter should also be added.

---

## [Author Response]

[Editors’ note: the author responses to the first round of peer review follow.]

Reviewer #1:[…] Overall, I am enthusiastic about the work, but I have a number of concerns.– First, the discovery of the Gpt system is not exactly novel, the Mla complex, apparently the direct homolog, has been previously identified in E. coli (which the author acknowledge). However contrary to previous interpretations, Gpt is proposed to transport glycerolipids from the IM to the OM while Mla (f I understand correctly) was proposed to maintain lipid asymmetry in the OM, transporting lipids from the OM to the IM. I am not an expert in this field but because this creates a controversy, I am wondering whether the data presented is sufficient to make this claim. The conclusion mostly comes from the assay presented in Figure 7 showing that newly synthesized phospholipids accumulate at the IM in gpt mutants. However, I do not understand why the OM composition appears mostly unaffected in the same mutants in that assay?

It is important to note that the data presented in Figure 7 of the original paper (now Figure 5 in the revised manuscript) is not a representation of the absolute value of newly synthesized phospholipids. Instead, we are presenting on the y-axis the molar ratio of newly synthesized, C13-labeled glycerophospholipid (GPL) to unlabeled glycerophospholipid. The data is presented in this manner to limit the need for normalization to something such as membrane protein content, which might be argued to be unreliable given that these mutations have been shown to severely disrupt the outer membrane. However, we recognize that this manner of presentation may cause unnecessary confusion about the assay, and so we have added additional figures to show the relative amount of newly synthesized GPL in the inner and outer membranes of wild type and mutant *A. baumannii* in the hope that this will more clearly represent the data. As detailed below, the data clearly shows that in the *mla* mutants newly synthesized GPL accumulate on the inner membrane.

At any given time, as GPL are transported from the IM to the OM by any mechanism, *mla* or otherwise, the likelihood that a single GPL molecule which is transported from the IM to the OM will be labeled is proportional to the ratio of labeled to unlabeled GPL in the inner membrane. Although the OM ratio is similar in the mutants and wild type, what is significant is the discrepancy in the ratios between the inner and outer membranes. As the ratio of labeled to unlabeled increases in the IM over time, it should also increase at the same rate in the OM, provided anterograde transport between the membranes is not inhibited, as demonstrated by the results of these experiments in wild type. The significance of our result is that newly synthesized GPL accumulate in the IM at a faster rate than in the OM in our mutants, indicating that some mechanism of anterograde transport is inhibited in these strains. However, as we note in the discussion, transport to the OM is not completely abolished, as there are still labeled GPL that appear in the OM under these conditions. The fact that the ratio of labeled to unlabeled GPL in the OM of mutants increases can be explained by the existence of additional unexplained mechanisms of GPL transport, perhaps including a passive flow of GPL between membranes that may occur during cell division, combined with the fact that the likelihood of transporting a labeled GPL is much higher in the mutants strains due to the excess accumulation of labelled GPL on the inner membranes, giving them a much greater proportion of labeled GPL on the inner membrane after 60 minutes.

Also given that the authors have this assay in hand it should not be too complicated to test the E. coli mlA mutant and see if they get similar results? That would help solving this controversy.Bottom line, this part should be absolutely convincing for the paper to be accepted.

We appreciate the reviewer’s suggestion and agree that it would be interesting to repeat our experiments in *Enterobacteriaceae*. However, the focus of our study is mechanisms of outer membrane permeability in *Acinetobacter*, and as such we have written this manuscript in order to present our findings about the Mla system in that organism. While we have preliminary data that the function of the system may be similar in *Salmonella typhimurium,* we think that repeating this work in another organism would be beyond the scope of this manuscript, and might serve as the basis of another manuscript entirely. We would also like to point out that we have made efforts to explain that while we believe the system functions primarily for anterograde rather than retrograde transport of GPL, we have not eliminated the possibility that the system does play some role in the maintenance of outer membrane asymmetry. In particular, the precise molecular function of the system’s outer membrane component, MlaA, is not entirely clear, and it is possible to imagine it might be in some way facilitating both the maintenance of lipid asymmetry and the transfer of GPL from MlaC to the outer membrane. It is not our intention in putting forth this data to stoke controversy, but to publish what we have observed about the function of this system in *Acinetobacter*. Pursuing the characterization of Mla in *E. coli* could conceivably put our model in direct conflict with the current paradigm and necessitate an extremely high burden of proof requiring additional experiments that could further delay publication of the data we already have.

– Some microscopy of the gpt mutants (EM, phase contrast) would be helpful to describe the phenotypes in these mutants. Could permeability of the mutants be shown directly in individual cells?

We have included in this revised manuscript phase contrast images of our mutants to help describe the phenotypes (Figure 1—figure supplement 1). It is our opinion that specific dye uptake or other visual methods would not add much over the quantitative assays of outer membrane permeability and antibiotic susceptibility we have performed.

Reviewer #2:[…] 1) Cryo-EM structural information of the GptBDEF IM complex. This is a novel structure that is only validated by the fitting of Phyre model structures of the cytoplasmic components (GptBF) and previous data showing that MlaD (GptD) is hexameric. Because the structure is of low-resolution (~11 Å), its impact is minimal – it does not provide insight into the mode of function or guidance for future work on function.

The overall features of both structures, solved independently, are identical, suggesting that they correspond to the correct structure for the complex. However, the limited resolution of the ecMlaBDEF complex structure did not allow modeling of its individual subunits, in contrast to the abMlaBDEF structure reported here. We have been able to further improve the resolution to 8.7 Å in this revised manuscript. Based on the modeling we saw more detail, which was not shown from the ecMlaBDEF structure. Similar to MlaB and MlaF proteins, most helices are well resolved, which allowed us to place the models unambiguously, the arrangement of MlaF clearly resembles the pre-translocation state of MalK. This suggests that we have trapped a similar conformation of the abMlaBDEF complex. We also show that while the periplasmic domain possesses 6-fold symmetry, the TM domains of MlaD do not appear symmetrical. Two of the TM domains of MlaD form close contacts with the density attributed to MlaE while the other four do not appear to contact any other proteins. We believe that this data will in fact prove useful to guide future work regarding the molecular function of the system.

2) The authors conclude that Gpt transports PLs from the IM to OM. This is very surprising because both the Mla system of E. coli and its chloroplast orthologous system have been reported to function in retrograde PL transport based on in vivo data. Previously, Malinverni and Silhavy (reference #8) concluded that Mla functions in retrograde transport because they obtained in vivo data demonstrating that PLs accumulate in the outer leaflet of the OM: in mla mutants, PagP was activated and defects were suppressed by PldA overexpression. It is impossible to explain those data in a model where Mla transports PLs from the IM to the OM, as it is suggested here for Gpt. It is also very difficult to imagine how a multi-protein transport system that is driven by an ABC transporter to transport PLs from one membrane to another could have evolved to run in opposite directions in different organisms.

As the reviewer describes, Malinverni and Silhavy demonstrated that mutations in the Mla system lead to activation of PagP, and PldA overexpression suppresses both the PagP activation phenotype as well as increased sensitivity observed in these mutants to SDS-EDTA, suggesting the accumulation of GPL in the outer leaflet of the outer membrane under those conditions. However, we strongly disagree with the reviewer’s statement that it is impossible to reconcile these observations with an anterograde transport model. As we note in our manuscript, conditions that perturb the outer membrane, such as chemical disruption via EDTA exposure, have been shown to result in an increase in membrane disorder, i.e. a reduction in membrane asymmetry following the spontaneous flipping of glycerophospholipids to the outer leaflet and activation of PagP (see references 27-29). It is reasonable therefore to suppose that mutation in *any* protein that contributes significantly to membrane barrier function might result in GPL exposure on the outer leaflet, and it does not necessarily mean that such a protein is directly responsible for maintaining lipid asymmetry under wild type conditions. Pulse-chase experiments have been performed to demonstrate that the orthologous system in the plant chloroplast functions to transport GPL originating in the endoplasmic reticulum towards the interior of the chloroplast for eventual delivery to the thylakoid membrane within the chloroplast (see reference 36). The Gram-negative bacteria we are discussing here are not contained within a eukaryotic cell exposed to organelles such as the ER that might be a source for GPL. Therefore, we do not find it difficult to imagine that this orthologous system might have evolved to have reverse functionality in the chloroplast from what we observe here in *A. baumannii.*

Here, Kamischke et al. conclude that Gpt is involved in anterograde transport because of data presented in Figures 6 and 7, where the authors measure PL content in the IM and OM. These data rely on the separation of IM and OM using a density ultracentrifugation procedure. How well these two membranes were separated is crucial to validate this work. Therefore, the authors should show the data they mentioned in the Materials and methods that illustrate proper separation of the IM and OM. In fact, the authors should at the very least show stained protein gels (not just immunoblots) of all fractions so that readers can assess the fractionation procedure.

In addition to the immunoblots, we have here included a figure of stained protein gels of inner and outer membrane fractions in both mutant and wild type bacteria so that readers can better assess the fractionation procedure.

Furthermore, I worry that this technique might have misled the authors because of the following issue. Separation of IM and OM only works when the density values of these membranes are different enough. There are two factors contributing to this difference: the different protein:lipid ratio between the two membranes and the LOS:phospholipid ratio in the OM. If the Gpt system was responsible for retrograde transport, as previously proposed for Mla, a gpt mutant would accumulate PLs in the OM. This would cause the density of the OM (or domains/areas in the OM) to be more IM-like, making the fractionation more difficult or even impossible depending on the extent of the defect. In fact, it is likely that if PLs accumulated in the OM, the OM would be a mixture of two types of bilayers: some areas would be like wild-type OM (asymmetric bilayer containing both LOS and PLs), while other areas would be IM-like (symmetric PL bilayer). That is, some OM domains (or areas) would fractionate as expected (heavy density), while others would fractionate with the IM or somewhere in between the OM and IM fraction (lighter density). If this happened, fractionations would appear normal except that one would expect excess PLs in the IM, compared to the OM. This is what the authors' data show, so I wonder if they are misinterpreting the data. Can the authors rule out this possibility?

The reviewer is correct to point out that areas of symmetric GPL bilayer in the OM might conceivably fractionate with the IM upon density centrifugation, leading to the appearance of an excess of GPL in the IM relative to the OM. This is one of the reasons we developed the stable isotope assay after first observing the overall reduction in OM GPL by TLC and LC-MS/MS shown in Figure 4. Additionally, it occurred to us that activation of OM phospholipases such as PldA could result in this phenotype and lead to mischaracterization of the system. The stable isotope assay gives insight into whether these results are due simply to mislocalization or degradation of OM GPL, or if they can in fact be attributed to deficient anterograde GPL transport. This is because the peak intensity of each newly synthesized, C13-labeled GPL is normalized to the corresponding unlabeled version of that GPL species with each sample injected into the LC-MS/MS. The result is a molar ratio of labeled to unlabeled GPL for every membrane sample. With C13-acetate as the sole carbon source, we observe a gradual increase in the ratio of labeled GPL relative to unlabeled GPL over time. In wild type bacteria, these ratios for the inner and outer membranes track closely over time, which indicates that under these conditions GPL transport from the inner to the outer membrane occurs quite rapidly, as has been previously shown in *E. coli* reference. In *mla* mutants the ratio is both higher and typically increases at a greater rate in the inner membrane. Phospholipases will not distinguish between labeled and unlabeled GPL species, and so will not impact the ratios obtained with this assay. Areas of symmetric OM bilayers fractionating with the IM also cannot explain the data for our mutants because that would not result in a specific increase in labeled GPL relative to unlabelled GPL, as such theoretical micro domains would themselves be a mixture of labeled and unlabelled GPL. The results observed in *mla* mutants are best explained as demonstrating a decreased rate of anterograde transport relative to wild type bacteria, in which GPL are first synthesized and inserted in the IM but subsequent transport to the OM is inhibited.

Furthermore, the authors use a fractionation procedure that is based on spheroplasting cells, which leaves domains/areas of the OM still connected to the IM via trans-envelope complexes (e. g. Tefsen J Biol Chem. 2005). Would the same results be obtained if cells were lysed by high pressure?

As described in our Materials and methods, following the spheroplast procedure, samples were homogenized under high pressure with an Avestin EmulsiFlex-C3 and spun down at 17,000 g for 10 min to removed un-lysed cells prior to ultracentrifugation.

Another issue that their data raises is this: The OM should cover more surface area than the IM. Here, the authors say that the OM of a gpt mutant has a lot less PLs than the OM of the wild type and that the IM of the mutant has more PLs than that of the wild type. Their data suggest that the OM in the gpt mutant covers less surface area that in IM. How is this possible? What is filling the void in the OM caused by the decrease in PLs? The authors do not provide any information about the overall composition of the OM in gpt mutants that could explain how it can have less PLs while the IM has more. Again, different composition of the OM could affect how it (or domains of it) fractionates.

Respectfully, we do not state that the IM of the *mla* mutants have more GPL than wild type, and our TLC and LC-MS/MS quantification data in fact show that IM GPL levels are comparable or slightly decreased in *mla* mutants. While our TLC and initial quantification of GPL by LCMS/ MS in Figure 4 suggest a decrease in OM GPL, we acknowledge in our manuscript that these results could be due to indirect effects, which is why we developed the stable isotope assay, as described in our response above. That being said, these mutants were revealed to us in a screen for outer membrane integrity, and have been shown to have an outer membrane permeability defect, suggesting that damage to the outer membrane is not fully compensated. Previous work implicating orthologous systems in GPL transport, and our own data showing GPL binding to components of the system led us to focus on the GPL components of the membranes rather than attempting to profile the overall membrane composition. We have not attempted to determine whether the LOS or lipoprotein profiles are significantly altered in these mutants so as to compensate for the loss of GPL, although this might be an interesting avenue for future study.

Regarding this question, our data showing a significant increase in exopolysaccharide in *mla* mutants may suggest regulatory mechanisms responding to membrane damage to mitigate the effects.

The authors should also address these additional major problems:3) Why isn't Figure 6B showing accumulation of PG and PE in the IM? These data seem to contradict the model. Even Figure 6A does not appear to show much of an increase of PLs at the IM.

It is likely that the inner membrane content of GPL is regulated by many factors including synthesis, degradation by phospholipases, outer membrane transport, and cell division. We can speculate that feedback mechanisms have resulted in maintenance of appropriate content of the inner membrane, which is critical to cellular integrity.

4) The authors state (without providing data) that cardiolipin is made at the OM. If that is true, based on the authors' model (Gpt transports the substrates needed to make cardiolipin at the OM), one would expect that the overall cardiolipin content in gpt mutants would be very low, especially in comparison to the other PL species. That is not what Figure 6A shows. Why?

Bioinformatic analysis of cardiolipin synthase genes in ATCC 17978 reveals two genes, one of which encodes a signal sequence for export to the periplasm. We do observe a reduction in OM CL by TLC as expected, however our stable isotope assay implicated the Mla system in anterograde transport of PE and PG, but not CL, indicating to us that there may be alternative mechanisms of increasing OM CL content, and we suggest that a periplasmic OM CL may explain this result, although we have not yet thoroughly investigated this as our focus is on the Mla system. We do not understand why the reviewer expects that the overall cardiolipin content in mutants would be low.

5) Could the authors please show error bars for Figure 7?6) Discussion section: "While it is possible that the Gpt system in E. coli serves a different primary function than in A. baumannii, the phenotypes observed in E. coli mla mutants might also be explained by a disruption of outer membrane structure stemming from decreased concentrations of outer membrane PLs, leading to activation of the PagP enzyme." This weak argument does not make any sense. PagP is activated by an increase in PLs at the outer leaflet of the OM.

We respectfully disagree. PagP is activated by OM damage such as the use of EDTA, which may or may not cause migration or membrane disorder. Despite the “dogma” that PagP is activated on migration of phospholipids to the outer membrane (which comes from a detergent based crystal structure) it is activated in the absence of any membrane damage in *Salmonella typhimurium*. The idea that PagP is activated on movement of membrane asymmetry is reasonable but also not necessarily the only mechanism. It is also plausible that a membrane defect has resulted in membrane disorder that allows PagP access to substrate in the inner leaflet.

Reviewer #3:The article is written and presented in a seamless manner, where figures, hypotheses, and figures are clearly articulated. This is a testament to the overall quality of the work and presentation. However, this is not to say that it has no weaknesses, some of which need to be addressed prior to publication. The authors' conclusions are never heavy-handed or written in absolute terms; as such they accommodate alternative opinions and possibilities; especially important, as their final figure, serves to upturn the existing dogma in the field, regarding the function of a homologous operon best studied in E. coli. Some might argue that renaming these homologs is unnecessary. The authors should strongly consider their proposal to rename these genes for the following reasons: i) the buried mention (Materials and methods) that cardiolipin is not transported by this system, argues that it is not in fact a general "glycerophospholipid" transporter (see below additional comments on cardiolipin); ii) these authors have only demonstrated in one strain of A. baumannii that some degree of PE and PG transport to the OM likely occurs as a consequence of this operon, and have not overturned the likely function of this operon to maintain asymmetry of the outer membrane. It's possible that context, such as organism or environments, favors utilization of this machinery in one direction or another. Also of great importance is the issue as to what fills in for the loss of GPL in the outer membrane as it is highly unlikely that it is LOS at the inner leaflet of the outer membrane. The authors' consideration on these matters, and their ability to adequately address these concerns will likely impact publication.

We thank the reviewer and agree with their points regarding renaming the system and acknowledge that we have not ruled out a role for this system in maintaining OM lipid asymmetry. In this version of the manuscript we will refer to the system as Mla.

Major concerns/comments:1) As noted above the Mla (Gpt) system has been previously characterized in E. coli and hypothesized as a retrograde GPL transporter. Clearly the mla system is not required for growth, so if it is functioning as an anterograde transporter what is replacing GPLs in the inner leaflet of the OM in mla mutants. In the view of this reviewer, the burden of proof is now higher given the previous work on the E. coli mla system.

Please see our response above to the similar concerns of reviewer #2 regarding the composition of the OM in *mla* mutants.

2) The audience must be provided with inner membrane and outer membrane separation data throughout the manuscript. This is critical as the reader must be convinced that changing the ratio of GPL/LOS/OM protein allows for proper separation of Acinetobacter membranes.

These controls have been included in the new version of the manuscript.

3) Do the authors have data to demonstrate that the K55L "dominant negative" mutation truly abrogates ATP hydrolysis, since it is a critical assumption in interpreting the data in Figures 1 and 2?

We have not attempted to directly demonstrate that the K55L mutation abrogates ATP hydrolysis. The Walker box lysine residue is highly conserved in ATPases and has been previously demonstrated in other ATPases to abrogate ATP hydrolysis (See references 15-18).

4) Could the authors please speculate why only A1S_3102 and A1S_3103 were identified using BCIP-Toluidine screen of Tn library? Where there not Tn insertions in the other members of the operon?

It is likely our screen was not performed to completion and that subsequent rounds of transposon mutagenesis would reveal additional Tn insertions in other components of the operon.

5) Why do the authors not discuss MlaA and why was a mutant not made in this gene? Also, it is not labeled in the model? Does it not function in the pathway in Acinetobacter?

Following our initial submission we have mutated and analyzed MlaA and observed results consistent with mutations of other components in the system. That data is now included in the manuscript.

6) Figure 6. Several comments on this figure:a) In 6A it appears there are more GPLs by TLC in the OM of the wild type which is odd.

We do not fully understand what the reviewer is referring to here with this observation.

b) In 6B, if gpt was a anterograde transporter one would expect to see accumulation of GPLs in the inner membrane and decreases of GPLs in the outer membrane. When you look at the bar graph, the levels of PE and PG don't increase in the inner membrane in the gptC mutant and they actually decrease a little. Why?

Gram-negative bacteria likely have multiple redundant mechanisms to ensure that the overall quality of IM GPL does not exceed a viable threshold relative to the OM. We reiterate our response to a similar concern raised by reviewer #2. It is likely that the inner membrane content of GPL is regulated by many factors including synthesis, degradation by phospholipases, outer membrane transport, and cell division. We can speculate that feedback mechanisms have resulted in maintenance of appropriate content of the inner membrane, which is critical to cellular integrity.

7) Have the authors performed electron microscopy to see if they observe changes in the inner membrane (membrane invaginations) because of loss of gpt mediated GPL transport? This sort of data would greatly strengthen their hypothesis.

This is an interesting idea but the lack of dramatic increase in GPL in the inner membrane has discouraged us from pursuing this line of research.

8) Discussion paragraph two: The Mla system was previously reported as a retrograde transport system for E. coli and in mla mutants PagP is activated. Since PagP is only functional when PLs are present at the surface, the idea was that in mla mutants GPLs are mislocalized in the outer leaflet of the OM. In the discussion, the authors rationalize that the phenotypes observed in E. coli mla mutants might also occur because GPL transport to the OM is not as efficient and because of this the LPS-to-GPL rations are off, resulting in flipping of GPLs to the outer leaflet and this is why PagP is activated. Is this likely, as PagP needs GPLs for activity and in the authors' model the Gpt (Mla) system is responsible, at least in part, for transport of GPLs to the OM?

We are not certain that PagP is only active on GPL in the outer leaflet. However there may be migration of GPL to the outer leaflet in the setting of the outer membrane defect of Mla mutants. It is possible even that MlaA could function to maintain lipid asymmetry by binding GPL in the outer membrane and presenting them to phospholipases and it could function as a receiver from MlaC to transport to the outer membrane. What seems unlikely is that there is retrograde transport of the GPL back to the inner membrane. PagP and phospholipases may function and the system may also transport from the inner membrane to the outer membrane.

9) Discussion paragraph three: The authors note that since the Mla (Gpt) system is upregulated in LOS deficient Acinetobacter, it suggests that Mla serves as a anterograde transporter as increased transport of GPLs may serve to compensate for lack of LOS in the outer membrane. Why couldn't it be the reverse, that the Mla system is upregulated in order to prevent too many GPLs in the OM. This goes back to what fills in the gaps when LOS is missing. After searching on PubMed, this reviewer found a recent paper from Boll et al., reporting that key lipoproteins seemed to be present in the outer leaflet of LOS deficient Acinetobacter.

The reviewer is correct to point out that the fact of Mla upregulation in LOS deficient *Acinetobacter* could be argued to support either a retrograde or an anterograde transport model. As such this point may not be useful for readers and we have removed it from the manuscript.

[Editors’ note: what follows is the authors’ plan to address the revisions.]

The current manuscript proposes that the Acinetobacter baumannii Mla system transports glycerophospholipid to the outer membrane. This statement challenges a large body of literature on both the E. coli and A. Baumannii systems (and to some extent V. Cholerae and H. influenza), all of which favor that Mla is instead involved in OM turnover and retrograde transport. After consultation with the reviewers, several key points would HAVE to be clarified for publication in eLife:1) The reference to existing literature must be improved in particular to reconcile and potentially explain the differences between this study and published studies. If needed, additional experiments must be provided to resolve/address these conflicts. Along these lines:– Powers and Trent, 2018, published evidence for retrograde transport in the same organism, which is not commented by the authors.– Contrarily to this study, multiple laboratories have reported that deletion of mlA genes is not associated with increased sensitivity to vancomycin and in fact linked to vancomycin resistance gene induction.– Roier et al. (Nat Comm, 2016) have shown that mlA deletion is linked to OMV production in V. Cholerae and H. influenza, which is not taken into account in this study. Can the authors rule out that lower GPL levels in the OM are not due to the loss of OM materials?2) The pulse chase assay is an important innovation of this study and the major experimental evidence to support the proposed directionality of Mla. Proving the consistency of this assay is therefore paramount to support the authors’ conclusions. The authors need to provide all the controls confirming that the IM and the OM were correctly separated in sucrose density gradients, which apparently is performed for the first time in A. Baumannii. For this, sucrose gradients need to be shown, with a complete set of IM and OM controls. A series of helpful recommendations is provided in the individual reviews. Variations in the assay starting point and in particular the high variance in total labeled lipids should also be explained.

We appreciate the opportunity to respond to the thoughtful comments of the reviewers and to share our plan to revise our manuscript to improve suitability for publication in *eLife*. We would like to thank the reviewers and editors for their time and careful discussions surrounding this work. Given the concerns of the editors and reviewers, we would like to offer the following clarifications of key points. The reviewers seemed to have three main points of concern for which we believe we can rapidly respond and facilitate the publication of our manuscript. The three concerns were 1) that our paper did not take into account existing literature on published phenotypes of *Acinetobacter baumannii* mutants; 2) that we did not provide adequate controls around membrane separation; and 3) about the initial labeling variability and potential alternative explanations of the radiolabeled lipid transport assay. In the case of the literature as discussed below in greater detail, we will take into account in the writing of the manuscript this additional paper that was published after we submitted the manuscript, though we do not believe our results are in conflict with it. As to membrane separation, we ran controls for outer and inner membrane separation that should have been included in the original manuscript, and they are very convincing in that they show that we are able to separate the membranes, and this should allay concerns in this regard. As to potential alternative explanations within the radiolabeled lipid assay (i.e., outer membrane blebbing or lipid hydrolysis), we discuss below how our choice to present data as a ratio of labeled to unlabeled lipid—rather than total lipid—controls for any loss of lipid through alternative methods. In short, while outer membrane blebbing or lipid breakdown may influence the total lipid content, it would not be expected to specifically target either labeled or unlabeled lipid and therefore would not affect the ratios presented. Further, we also discuss in detail below why the initial labeling may have variability but how it is similarly controlled by the measurement of ratios of labeled lipids. We hope this will allay the reviewers’ and editors’ concerns and allow us to publish this work, which is likely to be of interest to the field and stimulate discourse about this topic.

We intend to rewrite portions of the manuscript that refer to existing literature to further explain why we believe our hypothesis regarding the anterograde transport function of the Mla system is not disproven by existing literature. We are aware that our model of the Mla system conflicts with the dominant idea regarding the system’s function, and as such, our attempts to address previous work must be careful and deliberate. We intend to make every attempt in rewriting the manuscript to explain the limitations of our results, that they are in *Acinetobacter baumannii*, and that they do not rule out the possibility that under certain circumstance the system could additionally function to maintain outer membrane lipid asymmetry. However, we believe our conclusions regarding an anterograde transport function for the Mla system to be valid and supported by our data. We also intend to revise the manuscript to reflect the following:

1) The study by Powers and Trent was published in PNAS after our submission of this manuscript to *eLife*, and as such it has not been addressed in earlier versions. We intend to revise the manuscript to comment on this data, and to address those authors’ conclusions that their data reflect evidence of a retrograde transport function for Mla in *Acinetobacter baumannii*. Powers and Trent first obtained *A. baumannii* deficient in lipooligosaccharide (LOS) by selection in the presence of polymyxin B. They then performed an evolution experiment, passaging the strains in cultures containing polymyxin B over 120 generations, at which point they observed significantly improved growth in the populations. These evolved populations were also observed to have increased resistance to antibiotics including vancomycin, bacitracin, and daptomycin, and to appear more morphologically consistent relative to the unevolved strains when observed microscopically. Whole genome sequencing of the evolved strains revealed mutations in *mla* genes in seven of the 10 evolved populations. They also observed frequent disruptions in *pldA*, encoding an outer membrane phospholipase, as well as in other envelope genes. To further study these effects, they then introduced clean deletions of *mlaE* and *pldA* to ATCC 19606, and selected for LOS-deficient bacteria by plating on polymyxin B. These double mutants demonstrated improved growth and resistance to antibiotics, but continued to display altered cellular morphology. Mutations in *mlaE* in an LOS-deficient background were also shown by RNA-seq to be differentially regulated for 120 genes.

The authors present their data as evidence in support of Mla as a retrograde transport system. We would point out that lacking in their data is examination of the membrane glycerophospholipid (GPL) profile in their LOS-deficient mutants. It is assumed by the authors that *mla* and *pldA* mutations have the effect of stabilizing a symmetric outer membrane produced in the absence of LOS by allowing GPL to fill in the outer leaflet, resulting in improved growth and antibiotic resistance. Given that the data suggests that Mla and PldA are selected against when LOS is absent, examination of the outer membrane GPL content might have supported the authors’ conclusions if it revealed an increase in GPL in *mla* and *pldA* mutants. Absent such data, it is not obvious to us that the authors have sufficiently ruled out alternative explanations for their observed phenomena. For example, we would question the mechanisms regulating the homeostasis of both the inner and outer membranes and the entire periplasmic space in the absence of LOS. The authors acknowledge earlier work that observed an increase in expression of *mla* genes upon initial loss of LOS in 19606 (PMID: 22024825, PMID: 27681618). Genes in the *mla* pathway were shown to have an up to 7.5-fold increase in gene expression upon loss of LOS. Powers and Trent assert that the function of Mla is deleterious in the absence of LOS, but perhaps what is deleterious is the profound upregulation of *mla* expression in the absence of LOS, combined with an active PldA. If Mla is an anterograde transporter, we can imagine this might create a situation in which GPL are rapidly removed from the inner membrane and then degraded in the outer membrane in excess of what the cell can support, and limiting both of these processes together simply allows the cell to achieve a new homeostasis. The fact is, understanding of the myriad processes regulating bacterial outer membrane assembly and integrity remains limited even when LOS is present, and so interpreting results such as these is extremely difficult. Lastly, most data support a model in which the outer membrane is not a fluid membrane but an organelle assembled in chunks of proteins and lipids that are further assembled as a mosaic; and therefore it might be somewhat unpredictable whether adding GPL to pieces of the outer membrane without LOS would be detrimental or not, this idea is consistent with the large number of suppressor mutations the authors observed in the LOS mutant. Therefore, the authors’ interpretation of their data may not be as absolute as stated in their discussion, rather it in our view represents one possibility unsupported with biochemical data.

2) We are not aware of studies in *A. baumannii* that clearly refute our result of increased sensitivity to vancomycin in *mla* mutants. The reviewers did not indicate a specific reference in their comments. Powers and Trent observed an increase in vancomycin resistance in ATCC 19606 when both *mla* and *pldA* are mutated together, but the effect of either single mutation on vancomycin resistance is not shown. Recently, work performed in *Burkholderia* has shown that Bcc *mla* mutants are more sensitive to Gram-positive antibiotics such as macrolides and rifampin, as well as fluoroquinolones, tetracyclines, and chloramphenicol (PMID: 29986943), while those effects were not observed in *E. coli* or *Pseudomonas aeruginosa mla* mutants. This indicates that the effect of *mla* on antibiotic susceptibility may be species-dependent, and may differ in *A. baumannii* from other organisms that have been studied for reasons that are not yet known. In some organisms under the conditions grown in the laboratory, the amount of GPL transported and the expression of Mla may vary and reflect outer membrane integrity, perhaps leading to variability within species. However, we are confident in our results which have been repeated many times in multiple different independently isolated mutants.

3) The reviewers were interested in the idea that GPL could be lost from the surface by a mechanism that involved the formation of vesicles. While this could be true, it is not really relevant to our assay measuring GPL labeled ratios over time, which was developed in part because of this possibility. We considered the possibility of loss of GPL from the outer membrane due to outer membrane vesicle formation or more likely, by the possible increased activation of outer membrane phospholipases. Both would have the effect of removing GPL from the outer membrane and would result in lower GPL levels in the outer membrane of *mla* mutants. Neither of these mechanisms would operate specifically on either labeled or unlabeled lipids. We understood the decrease in outer membrane GPL to be insufficient evidence of an anterograde transport function for *mla,* and for this reason we developed the stable isotope assay to control for these possibilities. The stable isotope assay gives insight into whether these results are due simply to mislocalization or degradation of outer membrane GPL, or if they can in fact be attributed to deficient anterograde GPL transport. This is because the peak intensity of each newly synthesized, C13-labeled GPL is normalized to the corresponding unlabeled version of that GPL species with each sample injected into the LC-MS/MS. The result is a molar ratio of labeled to unlabeled GPL for every membrane sample. With C13-acetate as the sole carbon source, we observe a gradual increase in the ratio of labeled GPL relative to unlabeled GPL over time. In wild type bacteria, these ratios for the inner and outer membranes track closely over time, which indicates that under these conditions GPL transport from the inner to the outer membrane occurs quite rapidly, as has been previously shown in *E. coli* reference. In *mla* mutants the ratio is both higher and typically increases at a greater rate in the inner membrane. Phospholipases and budding outer membrane vesicles will not distinguish between labeled and unlabeled GPL species, and so will not impact the ratios obtained with this assay. We will include in our manuscript a discussion of OMV production in *mla* mutants and clarify that our stable isotope assay accounts for that possibility.

We appreciate the need to assure readers that the separation of the inner and outer membranes performed in this study is valid. We intend to provide enzymatic controls to further verify proper separation of the two membranes. We already have data that support the integrity of our samples using an NADH assay, as shown in Author Response Image 1. Our sucrose gradient contains three distinct concentrations of sucrose, and inner and outer membranes separate into distinct bands that are collected individually. To limit any potential mixing of the membranes, the inner membrane is collected from the top of the tube while the outer membrane is collected by puncturing the bottom of the tube and allowing the bottom band to be collected. We do not collect a gradient series as the reviewers may have seen performed by other laboratories. The image below may prove helpful for readers to understand this process and visualize the clear separation of *A. baumannii* membranes in this procedure. We can state that we visually observed no difference in membrane separation of *mla* mutants relative to wild type, but we did not record images of those membrane separations side by side. If those images would be helpful we can repeat the fractionation procedure to demonstrate.

The reviewers were concerned about some variability in the initial incorporation of label in the radiolabeled assay. We are aware that over the first hour of exposure, the initial rate at which C13-acetate was taken up and used to synthesize GPL appears to vary within a single strain over different experiments, as can be seen on examination of Supplementary file 2. We attribute this variability to minor differences in the optical density and growth time of the cultures following overnight growth with acetate as the sole carbon source, as well as to potentially slightly varying exposure to the stable isotope, although the exact reasons are not entirely clear. Although the starting values of isotope labeling varied, we consistently observed an increase in labeling in the inner membranes of *mla* mutants compared to the outer membrane that we did not observe in wild type, across multiple experiments. Comparing the relative percentage of isotope labeling on the inner membrane relative to the outer membrane in fact revealed consistent results across replicates, as we have presented in Figure 5C. The initial labeling differences do not necessarily reflect specific biological differences but represent a starting point for the assay. We present some raw data and not data normalized as many might submit because the initial labeling is not relevant and could be altered by the amounts of unlabeled acetate versus labeled in the culture.

[Editors’ notes: the authors’ response after being formally invited to submit a revised submission follows.]

After further consultation with the reviewers we decided that your response could adequately address the comments provided that in addition to the proposed changes the revision further contains the following new experiments:1) Given the existing literature, we would like further that the discrepancy on the vancomycin sensitivity be resolved by verification of this phenotype in a different A. baumannii strain.2) Completion of sucrose gradients with the proper controls to verify inner and outer membrane separation for both wild type and mla mutants (e.g. NADH oxidase assays). A clear description of the gradient methodology and images showing separation of the membrane as provided in the rebuttal letter should also be added.

We have rewritten portions of the manuscript that refer to existing literature to further explain why we believe our hypothesis regarding the anterograde transport function of the Mla system is not disproven by existing literature.

1) The study by Powers and Trent was published in PNAS after our submission of this manuscript to *eLife*, and as such it has not been addressed in earlier versions. We have revised the manuscript to comment on this data, and to address those authors’ conclusions that their data reflect evidence of a retrograde transport function for Mla in *Acinetobacter baumannii*.

2) Recently, work performed in *Burkholderia* has shown that Bcc *mla* mutants are more sensitive to Grampositive antibiotics such as macrolides and rifampin, as well as fluoroquinolones, tetracyclines, and chloramphenicol (PMID: 29986943), while those effects were not observed in *E. coli* or *Pseudomonas aeruginosa mla* mutants. This indicates that the effect of *mla* on antibiotic susceptibility may be species-dependent and may differ in *A. baumannii* from other organisms that have been studied for reasons that are not yet known. In some organisms under the conditions grown in the laboratory, the amount of GPL transported and the expression of Mla may vary and reflect outer membrane integrity. Given the reviewers’ concerns about the vancomycin sensitivity data, we repeated our experiments to test the sensitivity to vancomycin in another strain of *A. baumannii*, ATCC 5075. Comparison of wild type and *mlaF* deletion mutants in this strain did not show an increase in vancomycin sensitivity for *mlaF* mutants as we observed in ATCC 17978. In light of this, and to avoid confusion, we have removed the vancomycin MIC data from our revised manuscript.

3) We have included in our manuscript a discussion of OMV production in *mla* mutants and clarify that our stable isotope assay accounts for that possibility.

4) We appreciate the need to assure readers that the separation of the inner and outer membranes performed in this study is valid. We now provide enzymatic controls to further verify proper separation of the two membranes. Our data support the integrity of our samples using an NADH assay, as shown in Author Response Image 2. The image now provided in Figure 5—figure supplement 3 should prove helpful for readers to understand this process and visualize the clear separation of *A. baumannii* membranes in this procedure.

**Author response image 2. respfig2:**